# Off-policy learning in large action spaces: Optimization matters more than estimation

## Abstract

Off-policy evaluation (OPE) and off-policy learning (OPL) are foundational for decision-making in offline contextual bandits. Recent advances in OPL primarily optimize OPE estimators with improved statistical properties, assuming that better estimators inherently yield superior policies. Although theoretically justified, this estimator-centric approach neglects a critical practical obstacle: challenging optimization landscapes. In this paper, we provide theoretical insights and empirical evidence showing that current OPL methods encounter severe optimization issues, particularly as the action space grows. We show that estimator-aware policy parametrization can mitigate, but not fully resolve, optimization challenges. Building on this, we explore simpler weighted log-likelihood objectives and demonstrate that they enjoy substantially better optimization properties and still recover competitive, often superior, learned policies. Our findings emphasize the necessity of explicitly addressing optimization considerations in the development of OPL algorithms for large action spaces.

## 1 Introduction

The offline contextual bandit (Dudík et al., 2011) leverages logged data from past interactions to improve future decision-making, with wide applications in areas like recommendation systems (Bottou et al., 2013; Aouali et al., 2022). We consider a standard setting where we are given a dataset $\mathcal{D}_n = \{(X_i, A_i, R_i)\}_{i=1}^n$ of $n$ i.i.d. tuples. Each tuple consists of a context $X_i \in \mathcal{X} \subseteq \mathbb{R}^d$ drawn from an unknown distribution $\nu$, an action $A_i \in \mathcal{A} = [K]$ sampled from a known logging policy as $A_i \sim \pi_0(\cdot \mid X_i)$, and a corresponding reward $R_i \sim p(\cdot \mid X_i, A_i)$ sampled from the unknown reward distribution $p(\cdot \mid X_i, A_i)$, whose mean is $r(x, a) = \mathbb{E}_{R \sim p(\cdot \mid x, a)}[R]$. The performance of any new policy $\pi$ is measured by its *value*, defined as the expected reward it would obtain:

$$V(\pi) = \mathbb{E}_{X \sim \nu, A \sim \pi(\cdot \mid X)}[r(X, A)]. \tag{1}$$

The goal of *off-policy learning (OPL)* is to leverage the offline dataset $\mathcal{D}_n$ to learn a policy $\hat{\pi}_n$ from a policy class $\Pi$ that maximizes this value, i.e., $\hat{\pi}_n = \arg\max_{\pi \in \Pi} V(\pi)$.

The dominant paradigm in OPL is to optimize an *off-policy evaluation (OPE)* estimator $\hat{V}_n(\pi)$ that approximates the true policy value $V(\pi)$ (Swaminathan & Joachims, 2015a) such as $\hat{V}_n(\pi) \approx V(\pi)$. The learning problem is thus framed as $\hat{\pi}_n = \arg\max_\pi \hat{V}_n(\pi)$, with the rationale that maximizing a more accurate estimate of the value yields a superior learned policy. However, this estimator-centric view overlooks a critical aspect: the optimization landscape. OPE objectives (Dudík et al., 2011; Dudík et al., 2012; Dudik et al., 2014; Wang et al., 2017; Farajtabar et al., 2018; Su et al., 2020; Metelli et al., 2021; Kuzborskij et al., 2021; Saito & Joachims, 2022) are highly non-concave with common parameterized policies (Chen et al., 2019), prone to suboptimal local maxima, an issue more pronounced in large action spaces. Notably, even sophisticated estimators designed to reduce variance fail to overcome this optimization barrier, suffering from difficult-to-optimize landscapes.

We show that one way to alleviate these difficulties is through *estimator-aware policy parametrization*: structuring the policy class to match the implicit biases of the estimator. Such parametrizations reduce the effective search space and can shorten optimization plateaus. While this strategy provides tangible benefits, it does not eliminate the fundamental non-concavity of OPE objectives, leaving optimization as the central bottleneck.

Motivated by this limitation, our work advocates an alternative approach based on *policy-weighted log-likelihood (PWLL)* objectives. Unlike traditional estimators, PWLL optimizes an objective $\hat{U}_n(\pi)$ designed for ease of optimization rather than accuracy in estimating $V(\pi)$. Although PWLL objectives perform poorly as value estimators, their favorable concave landscape significantly enhances their effectiveness for learning. Through theoretical and empirical analysis, we show that this optimization-centric approach consistently enables simpler PWLL objectives to outperform complex, state-of-the-art OPE objectives, particularly in large action spaces.

## 2 ANALYSIS OF OPE OBJECTIVES

OPE objectives optimize an estimator $\hat{V}_n(\pi)$ of the value $V(\pi)$. While statistically motivated, this induces objective-specific biases in the learned policy and yields challenging optimization landscapes. To make this explicit, we study the *asymptotic solutions* $\pi_*^{\text{METHOD}} = \operatorname{argmax}_\pi \lim_{n\to\infty} \hat{V}_n^{\text{METHOD}}(\pi)$ obtained by maximizing each estimator in the infinite-data regime (proofs in Appendix C).

**Why is the infinite-data view informative?** In practice, the logging policy $\pi_0$ typically concentrates on a small, context-dependent support (Sachdeva et al., 2020) that is much smaller than $K$:

$$S_0(x) = \{a : \pi_0(a \mid x) > 0\}, \qquad k_0(x) = |S_0(x)| \ll K, \qquad (2)$$

and we denote $k_0 = \mathbb{E}[k_0(X)]$ as the typical support size. For a fixed policy $\pi$, standard concentration results (e.g., (Sakhi et al., 2024)) give pointwise deviations of order $\sqrt{k_0/n}$ between $\hat{V}_n(\pi)$ and its expectation.[1] Hence, when $n$ is large and $k_0$ is small (common in practice), the estimator is close to its value with infinite data (i.e., its expectation). Therefore, looking at the maximizer of the estimator in the infinite-data regime (i.e., asymptotic solution) is insightful. In addition, these asymptotic solutions could be obtained in closed-form, while the finite-data ones cannot. This is why we focus on the infinite-data view.

**What does the infinite-data view reveal?** Taking $n \to \infty$ removes sampling fluctuations and isolates the inductive bias of the objective. Different estimators converge to different policy structures even with infinite data. Thus, $\pi_*^{\text{METHOD}}$ is governed by the estimator's design rather than by statistical noise. This perspective clarifies the policies each objective targets and motivates *objective-aware parametrizations*, aligning the policy class with the induced bias, *to ease optimization*. This forms the first line of improvements we propose in this paper.

### 2.1 STANDARD OPE OBJECTIVES

**Inverse propensity scoring (`IPS`).** The foundational `IPS` estimator (Horvitz & Thompson, 1952) re-weights observed rewards by the ratio between the target policy $\pi$ and the logging policy $\pi_0$:

$$\hat{V}_n^{\text{IPS}}(\pi) = \frac{1}{n} \sum_{i=1}^n \frac{\pi(A_i \mid X_i)}{\pi_0(A_i \mid X_i)} R_i. \qquad (3)$$

With infinite data, `IPS` selects the best-rewarding action among those in the support of $\pi_0$:

$$\pi_*^{\text{IPS}}(a \mid x) = \mathbb{1}\left[a = \operatorname*{argmax}_{a' \in \mathcal{A}} r(x, a')\mathbb{1}[\pi_0(a' \mid x) > 0]\right]. \qquad (4)$$

`IPS` is unbiased but can suffer from high variance due to large importance weight values.

**Clipped IPS (`cIPS`).** To mitigate the high variance of `IPS`, `cIPS` (Bottou et al., 2013) clips small propensity scores at a threshold $\tau \in (0, 1)$:

$$\hat{V}_n^{\text{cIPS}}(\pi) = \frac{1}{n} \sum_{i=1}^n \frac{\pi(A_i \mid X_i)}{\max\{\pi_0(A_i \mid X_i), \tau\}} R_i. \qquad (5)$$

This clipping introduces a bias. The asymptotic policy down-weights the rewards of rare actions, causing it to favor actions that were frequent under $\pi_0$, even if they are suboptimal:

$$\pi_*^{\text{cIPS}}(a \mid x) = \mathbb{1}\left[a = \operatorname*{argmax}_{a' \in \mathcal{A}} \frac{\pi_0(a' \mid x)}{\max\{\pi_0(a' \mid x), \tau\}} r(x, a')\right]. \qquad (6)$$

---

[1]This can be extended to hold uniformly over a parametric class of dimension $d$ (e.g., linear softmax), via standard Rademacher/VC arguments.

**Exponential smoothing (`ES`).** Instead of hard clipping, `ES` (Aouali et al., 2023) smooths importance weights by raising propensities to a fractional power $\alpha \in (0, 1)$:

$$\hat{V}_n^{\text{ES}}(\pi) = \frac{1}{n} \sum_{i=1}^n \frac{\pi(A_i \mid X_i)}{\pi_0(A_i \mid X_i)^\alpha} R_i. \tag{7}$$

Its asymptotic policy balances reward maximization with preference for frequent actions:

$$\pi_*^{\text{ES}}(a \mid x) = \mathbb{1}\left[ a = \operatorname*{argmax}_{a' \in \mathcal{A}} r(x, a')\pi_0(a' \mid x)^{1-\alpha} \right]. \tag{8}$$

Another variant of `ES` regularizes the entire importance weight as $(\frac{\pi}{\pi_0})^\alpha$ instead of only the denominator. In contrast to the deterministic policies derived from `IPS`, `cIPS`, and the `ES` formulation above, this approach yields a stochastic asymptotic policy: $\pi_*^{\text{ES}}(a \mid x) \propto r(x,a)^{1/(1-\alpha)}\pi_0(a \mid x)$. Other regularizations include logarithmic smoothing (Sakhi et al., 2024), implicit exploration (Gabbianelli et al., 2024), harmonic correction (Metelli et al., 2021), shrinkage (Su et al., 2020). Further details are available in Appendix B.

**Doubly robust (`DR`).** The `DR` estimator incorporates a reward model $\hat{r}(x, a)$ to reduce variance and enable generalization to actions outside $\pi_0$'s support. A common clipped variant is:

$$\hat{V}_n^{\text{DR}}(\pi) = \frac{1}{n} \sum_{i=1}^n \frac{\pi(A_i \mid X_i)}{\max\{\pi_0(A_i \mid X_i), \tau\}} \left(R_i - \hat{r}(X_i, A_i)\right) + \mathbb{E}_{A \sim \pi(\cdot \mid X_i)}\left[\hat{r}(X_i, A)\right]. \tag{9}$$

Its solution interpolates between model-based reward prediction and unbiased correction using $\pi_0$:

$$\pi_*^{\text{DR}}(a \mid x) = \mathbb{1}\left[ a = \operatorname*{argmax}_{a' \in \mathcal{A}} \hat{r}(x, a') + \frac{\pi_0(a' \mid x)}{\max\{\pi_0(a' \mid x), \tau\}} \left(r(x, a') - \hat{r}(x, a')\right) \right]. \tag{10}$$

## 2.2 Large-scale OPE objectives

In large action spaces, importance weights $\frac{\pi(a \mid x)}{\pi_0(a \mid x)}$ can become huge, leading to estimators with high variance. To mitigate this, modern methods compute marginalized importance weights over a lower-dimensional action representation, trading bias for reduced variance.

**Marginalized IPS (`MIPS`).** `MIPS` (Saito & Joachims, 2022) tackles large action spaces by clustering actions. It maps each action $a$ to a cluster $c$ via a function $h : \mathcal{A} \to \mathcal{C}$, where $|\mathcal{C}| \ll |\mathcal{A}|$. Estimation is then performed at the cluster level:

$$\hat{V}_n^{\text{MIPS}}(\pi) = \frac{1}{n} \sum_{i=1}^n \frac{\pi(C_i \mid X_i)}{\pi_0(C_i \mid X_i)} R_i, \quad \text{where } C_i = h(A_i) \text{ and } \pi(c \mid x) = \sum_{a \in c} \pi(a \mid x). \tag{11}$$

This cluster-level marginalization introduces bias: the asymptotic solution only selects the best *cluster* based on its average reward under $\pi_0$, and cannot differentiate between actions within that cluster:

$$\pi_*^{\text{MIPS}}(c \mid x) = \mathbb{I}\left[ c = \operatorname*{argmax}_{c' \in \mathcal{C}} \left\{ \frac{\sum_{a \in c'} \pi_0(a \mid x)r(x, a)}{\sum_{a \in c'} \pi_0(a \mid x)} \right\} \right]. \tag{12}$$

Hence, `MIPS` offers no specific guidance for selecting an action within the optimal cluster; any action is considered equally valid. Consequently, if actions are chosen uniformly at random from this optimal cluster, the resulting asymptotic action-level solution is:

$$\pi_*^{\text{MIPS}}(a \mid x) = \frac{\mathbb{I}\left[ h(a) = \operatorname{argmax}_{c' \in \mathcal{C}} \left\{ \frac{\sum_{a \in c'} \pi_0(a \mid x)r(x, a)}{\sum_{a \in c'} \pi_0(a \mid x)} \right\} \right]}{|h(a)|}.$$

where $|h(a)|$ denotes the size of the cluster containing action $a$.

**Conjunct effect modeling (`OffCEM`).** Building on `MIPS`, `OffCEM` (Saito et al., 2023) uses a reward model $\hat{r}$ to correct for the cluster-level aggregation bias, in a doubly robust fashion:

$$\hat{V}_n^{\text{OffCEM}}(\pi) = \frac{1}{n} \sum_{i=1}^n \left( \frac{\pi(C_i \mid X_i)}{\pi_0(C_i \mid X_i)} \left(R_i - \hat{r}(X_i, A_i)\right) + \mathbb{E}_{A \sim \pi(\cdot \mid X_i)}[\hat{r}(X_i, A)] \right). \tag{13}$$

The resulting asymptotic policy selects the action that maximizes the model-predicted reward $\hat{r}$, plus a cluster-level correction term that accounts for model error:

$$\pi_*^{\text{OffCEM}}(a \mid x) = \mathbb{I}\left[a = \underset{a' \in \mathcal{A}}{\operatorname{argmax}}\left\{\hat{r}(x, a') + \frac{\sum_{\bar{a} \in h(a')} \pi_0(\bar{a} \mid x)(r(x, \bar{a}) - \hat{r}(x, \bar{a}))}{\sum_{\bar{a} \in h(a')} \pi_0(\bar{a} \mid x)}\right\}\right]. \quad (14)$$

**Two-stage decomposition (POTEC).** In this work, we see POTEC (Saito et al., 2025) as an *optimization strategy of* OffCEM (rather than seeing it as a new estimator). It restricts the policy to a cluster-informed form,

$$\pi(a \mid x) = \sum_{c \in \mathcal{C}} \pi^{\text{RM}}(a \mid x, c)\pi^{\text{CL}}(c \mid x),$$

where $\pi^{\text{RM}}(a \mid x, c) = \mathbb{1}[a = \operatorname{argmax}_{a' \in c} \hat{r}(x, a')]$ is fixed, model-based policy that deterministically selects the best action within each cluster. Learning is then simplified to finding the optimal cluster-level policy $\pi^{\text{CL}}$ that maximizes the OffCEM objective in Eq. (13):

$$\hat{V}_n^{\text{POTEC}}(\pi^{\text{CL}}) = \frac{1}{n}\sum_{i=1}^{n}\left(\frac{\pi^{\text{CL}}(C_i \mid X_i)}{\pi_0(C_i \mid X_i)}(R_i - \hat{r}(X_i, A_i)) + \sum_{c \in \mathcal{C}}\pi^{\text{CL}}(c \mid X_i)\hat{r}_c^*(X_i)\right), \quad (15)$$

where $\hat{r}_c^*(x) = \max_{a \in c}\hat{r}(x, a)$ is the estimated reward of the best action in cluster $c$. This practical decomposition has the same optimal asymptotic solution as OffCEM: $\pi_*^{\text{POTEC}} = \pi_*^{\text{OffCEM}}$.

**Policy convolution (PC).** Moving beyond hard clustering, PC (Sachdeva et al., 2023) leverages the assumption that actions close in an embedding space yield similar rewards. For each action $a$, it aggregates over its neighborhood of nearest neighbors $N_\epsilon(a) = \{a' : d(a, a') < \epsilon\}$, where $d$ is a pre-defined distance metric (e.g., $\ell_2$ distance between embeddings):

$$\hat{V}_n^{\text{PC}}(\pi) = \frac{1}{n}\sum_{i=1}^{n}\frac{\pi(N_\epsilon(A_i) \mid X_i)}{\pi_0(N_\epsilon(A_i) \mid X_i)}R_i, \quad \text{with } \pi(N_\epsilon(a) \mid x) = \sum_{a' \in N_\epsilon(a)}\pi(a' \mid x). \quad (16)$$

The induced asymptotic policy is deterministic: it selects the action $a'$ that maximizes an aggregated neighborhood score. Each logged neighbor $\bar{a} \in N_\epsilon(a')$ contributes its reward $r(x, \bar{a})$, weighted by the conditional probability of observing $\bar{a}$ under the logging policy restricted to its neighborhood.

$$\pi_*^{\text{PC}}(a \mid x) = \mathbb{I}\left[a = \underset{a' \in \mathcal{A}}{\operatorname{argmax}}\left\{\sum_{\bar{a} \in N_\epsilon(a')}\frac{\pi_0(\bar{a} \mid x)r(x, \bar{a})}{\pi_0(N_\epsilon(\bar{a}) \mid x)}\right\}\right]. \quad (17)$$

Other recent IPS variants for large action spaces (Peng et al., 2023; Cief et al., 2024; Taufiq et al., 2024) are often extensions of MIPS that relax its core assumptions. We focused on four methods (MIPS, OffCEM, POTEC, and PC), which we consider representative of this family. Since these variants largely share the same MIPS foundation and optimization procedure (with the notable exception of POTEC), we expect our findings to be generally applicable.

## 2.3 OPTIMIZATION CHALLENGES

The statistical properties of OPE estimators are only half the story. In practice, their effectiveness is often limited by a more immediate obstacle: a challenging optimization landscape. OPE objectives become very difficult to optimize when paired with standard, expressive policy classes like the softmax. This section explores why this happens and introduces *objective-aware parametrization* as a strategy to mitigate, though not entirely solve, the problem.

To analyze the optimization process, we consider policies parametrized by a softmax function. For any given context $x$, the policy is defined over an *effective action space*, $\mathcal{A}_{\text{EFF}}(x) \subseteq \mathcal{A}$, which is the set of actions that can be assigned non-zero probability. The policy takes the form:

$$\pi_\theta(a|x) = \frac{\exp(s_\theta(x, a))}{\sum_{a' \in \mathcal{A}_{\text{EFF}}(x)}\exp(s_\theta(x, a'))} \quad \text{for } a \in \mathcal{A}_{\text{EFF}}(x), \quad (18)$$

where $s_\theta(x, a)$ is a learnable score function. A common choice are linear softmax scores,

$$\text{lightweight: } s_\theta(x, a) = h(x, a)^\top \theta, \qquad\qquad \text{heavyweight: } s_\theta(x, a) = h(x)^\top \theta_a, \qquad (19)$$

and we respectively call them, *lightweight parametrization* (learning a single shared parameter vector $\theta$) and *heavyweight parametrization* (learning separate parameters $\theta_a$ for each action).

The size of this effective action space, $K_{\text{EFF}}(x) = |\mathcal{A}_{\text{EFF}}(x)|$, is the critical factor governing optimization. The following propositions (proofs in Appendix D, adapted from (Chen et al., 2019; Mei et al., 2020a)) reveal just how severe the problem can be.

First, gradient-based methods can get stuck in suboptimal regions for extended periods.

**Proposition 2.1** (Optimization plateaus). *For any OPE estimator $\hat{V}_n$ that is linear in $\pi$, and even with a linear softmax policy, there exist problems where gradient descent is trapped in a suboptimal region for a number of iterations that scales linearly with the effective action space size, $K_{\text{EFF}}$.*

Second, the optimization landscape is highly non-concave, plagued by poor local optima that can trap the learning algorithm.

**Proposition 2.2** (Local maxima). *Under similar conditions, the optimization landscape for OPE objectives can have a number of local maxima that is exponential in $K_{\text{EFF}}$.*

These results highlight that $K_{\text{EFF}}$ plays a central role in the optimization properties of OPL. A common implementation choice is to set the effective action space to the full action space for all contexts, i.e., $\mathcal{A}_{\text{EFF}}(x) = \mathcal{A}$, which makes $K_{\text{EFF}}(x) = K$. In large-scale settings where $K$ can be in the millions, this is a recipe for optimization failure, as learning must navigate a landscape with potentially long plateaus and an exponentially large number of local maxima.

Surprisingly, even sophisticated methods designed specifically for large action spaces often fall into this trap. At first glance, methods like MIPS, OffCEM, and PC appear to operate in a smaller space because their objective functions involve marginalized probabilities like $\pi(C_i \mid X_i)$ in MIPS and OffCEM or $\pi(N_\epsilon(A_i) \mid X_i)$ in PC. However, these marginalized terms are defined as sums over an underlying action-level policy:

$$\pi(C_i \mid X_i) = \sum_{a \in C_i} \pi(a \mid X_i), \qquad \text{and} \qquad \pi(N_\epsilon(a) \mid x) = \sum_{a' \in N_\epsilon(a)} \pi(a' \mid x).$$

If this foundational policy, $\pi(a \mid x)$, is parameterized as a standard softmax over the entire action space $\mathcal{A}$, then the optimization procedure must still compute gradients with respect to the scores of all $K$ actions. The learning problem does not shrink; the complexity is merely hidden inside the definition of the cluster-level probabilities.

The only exception among these is POTEC, which we view not just as an estimator but as a deliberate *optimization strategy*. By fixing the intra-cluster policy $\pi^{\text{RM}}$ and learning only the cluster-level policy $\pi^{\text{CL}}$, POTEC fundamentally changes the problem. It forces the optimization to occur directly in the cluster space, making its effective action space $\mathcal{C}$ and its cardinality $|\mathcal{C}| \ll K$. This structural choice circumvents the bottleneck entirely, easing the optimization landscape.

### 2.3.1 DESIGN IMPLICATIONS: OBJECTIVE-AWARE PARAMETRIZATION

The choice of $K_{\text{EFF}}$ introduces a fundamental trade-off. A smaller effective action space leads to a simpler optimization landscape, it also risks excluding the optimal action and reduces the expressiveness of the policy class. If $\mathcal{A}_{\text{EFF}}(x)$ is chosen arbitrarily and is too restrictive, it may lead to bad performance. The challenge is to find the *sweet spot*: a parametrization that is constrained enough to be optimizable, yet expressive enough to contain the best possible policy for a given objective.

This is precisely where our asymptotic analysis helps. The asymptotic solution $\pi_*^{\text{METHOD}}$ for each estimator reveals the minimal sufficient set of actions required to find the objective's maximizer. By aligning the policy parametrization with this insight, we can shrink the search space without sacrificing performance. This is the core principle of *objective-aware parametrization*.

For instance, the asymptotic solutions for IPS, cIPS, ES are always confined to the support of the logging policy, $S_0(x)$. This tells us that setting $\mathcal{A}_{\text{EFF}}(x) = S_0(x)$ is a sufficient parametrization.

Similarly, for `OffCEM` and `MIPS`, the cluster-level structure of their asymptotic solutions suggests that a two-stage decomposition like `POTEC` is the most effective parametrization. These observations allow us to identify sufficient policy parametrizations that reduce the degrees of freedom and make learning more tractable. Based on this, we make the following claims, which we validate empirically in Section 4:

**Claim 2.3.** *For importance-weighted objectives (e.g., `IPS`, `cIPS`, `ES`), parameterizing the policy $\pi$ with support restricted to that of the logging policy, $S_0(x)$, improves optimization and leads to superior learned policies.*

**Claim 2.4.** *For large-scale objectives (e.g., `OffCEM`), using a two-stage decomposition (`POTEC`-style) that optimizes a policy at the cluster level yields better optimization and final performance than a naive action-level parametrization.*

While this objective-aware parametrization makes the optimization landscape of OPE objectives more navigable by shrinking the search space, it only treats the symptoms (plateaus and local maxima) without curing the underlying non-concavity. Thus, we propose next a more fundamental shift: abandoning accurate value estimation in favor of tractable optimization.

## 3 ANALYSIS OF PWLL OBJECTIVES

To overcome the optimization challenges of OPE objectives, we consider policy-weighted log-likelihood (PWLL) objectives. These methods trade accurate value estimation for a well-behaved, concave optimization landscape, leading to more robust and effective policy learning.

**General form.** Given a positive weighting function $g(r, p_0)$, the PWLL objective is:

$$\hat{U}_n^g(\pi) = \frac{1}{n} \sum_{i=1}^n g(R_i, \pi_0(A_i \mid X_i)) \log \pi(A_i \mid X_i). \tag{20}$$

The key motivation behind PWLL is to replace the linear dependence on the policy in OPE estimators, responsible for plateaus and local maxima in Section 2, with a concave transformation. Softmax policies are parametrized through scores $s_\theta(x, a)$, and the map $s \mapsto \log \text{softmax}(s)$ is concave. Consequently, for common linear parametrizations in Eq. (19), the composition $\log \pi_\theta(a \mid x)$ is concave in $\theta$. This removes the optimization pathologies inherent to OPE objectives. Proposition 3.1 (proof in Appendix D.) formalizes this advantage.

**Proposition 3.1.** *For linear softmax policies $\pi_\theta$, the PWLL objective $\hat{U}_n^g(\pi_\theta)$ is concave, and is strongly concave once $\ell_2$ regularization is used.*

Proposition 3.1 makes PWLL appealing for stochastic optimization. In Appendix E, we show that under standard assumptions of bounded feature norms $\|h(x, a)\|$ and weights $g(R_i, \pi_0(A_i, X_i))$, these objectives satisfy the regularity conditions necessary to invoke established convergence theorems (Garrigos & Gower, 2023). This allows us to derive problem-dependent convergence guarantees: stochastic gradient ascent attains a global $\mathcal{O}(1/\sqrt{T})$ rate in the general (concave) case (Proposition E.4), accelerating to a geometric rate under $\ell_2$-regularization (Proposition E.5). In contrast, such global guarantees cannot be derived for OPE objectives due to their inherent non-concavity.

Beyond optimization properties, PWLL also admits a simple statistical interpretation. $\hat{U}_n^g(\pi)$ in Eq. (20) is a weighted log-likelihood: the term $\log \pi(A_i \mid X_i)$ performs standard behavior cloning, while the weight $g(R_i, \pi_0(A_i \mid X_i))$ determines how *desirable*[2] each logged sample is. This turns off-policy learning into a form of logging-aware and reward-weighted maximum-likelihood estimation. Different choices of $g$ encode different notions of desirability. For example, the weighting

$$g(r, \pi_0(a \mid x)) = \frac{r}{\max\{\pi_0(a \mid x), \tau\}}$$

emphasizes samples with high reward while reducing the influence of actions that the logging policy selected very frequently. At the same time, the clipping at $\tau$ prevents extremely rare actions from receiving disproportionately large weights, ensuring that their contribution is attenuated once $\pi_0(a \mid$

---

[2]By how desirable an action is, we mean how strongly this action should influence the learned policy.

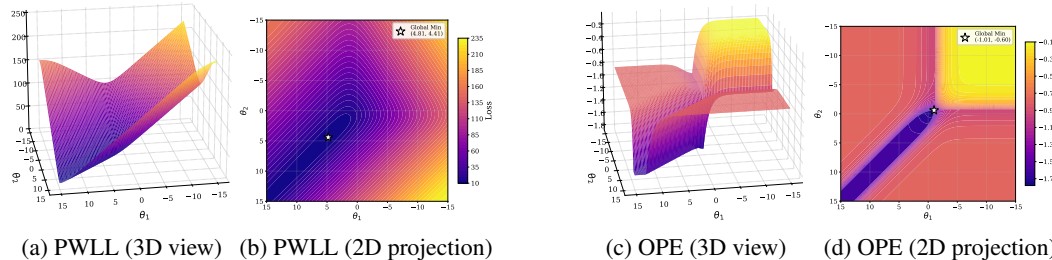

(a) PWLL (3D view)    (b) PWLL (2D projection)      (c) OPE (3D view)    (d) OPE (2D projection)

Figure 1: Optimization landscapes on a toy example. PWLL (`cLPI`) vs OPE (`cIPS`).

$x$) falls below the threshold. In this view, desirable samples are those that provide strong reward evidence without allowing very small propensities to dominate the updates. Many other PWLL variants arise from different choices of $g$ (see below), each specifying a distinct prioritization scheme for the logged data, while all benefit from the concavity induced by the logarithmic term.

To illustrate the qualitative difference between PWLL and OPE objectives, we construct a simple offline bandit problem with $K = 3$ actions and visualize the resulting optimization landscapes in a two-parameter policy space. Concretely, we consider a non-contextual setting with deterministic mean rewards $r = (0.9, 0.7, 0.2)$ and a logging policy $\pi_0$ whose support places almost all mass on action 3 ($\pi_0(1) = 0.002$, $\pi_0(2) = 0.003$, $\pi_0(3) = 0.995$). We generate a fixed dataset of $n = 60$ logged samples $(A_i, R_i)$ by drawing actions $A_i \sim \pi_0$ and binary rewards from the corresponding Bernoulli distributions, $R_i \sim \text{Bern}(r(A_i))$. To obtain a two-dimensional visualization, we parameterize the target policy using a softmax over three logits: $\pi_\theta(a) = e^{\theta_a} / \sum_{b \in [3]} e^{\theta_b}$, fixing the logit associated with action 3 as $\theta_3 = 1$, and letting the remaining two logits be free parameters $(\theta_1, \theta_2)$.

In Fig. 1, the PWLL landscape is concave with well-scaled gradients, and optimization trajectories converge reliably from roughly any initialization. In contrast, the OPE landscape consists of flat regions, separated by a narrow band of extremely steep curvature. This creates both vanishing and exploding gradients, severe ill-conditioning, and high sensitivity to initialization and learning rate. This aligns with the optimization pathologies in Propositions 2.1 and 2.2.

*Remark* 3.2 (Beyond linear-softmax policies). The concavity guarantee of Proposition 3.1 assumes linear-softmax policies. In many large-scale recommendation systems, a deep encoder is pre-trained and kept fixed, and only a final linear head is optimized for the downstream task; in this case, the policy is still linear-softmax in the trainable parameters, and PWLL objectives retain their concavity. When the full network is trained end-to-end, concavity no longer holds. Yet, PWLL's gradients $g(R_i, \pi_0(A_i|x_i))\nabla_\theta \log \pi_\theta(A_i \mid X_i)$ match the structure of cross-entropy gradients, which are known to produce stable and well-scaled updates in deep architectures. Thus, even without formal guarantees, PWLL maintains substantially more benign optimization dynamics than OPE objectives.

**Local policy improvement (`LPI`).** Liang & Vlassis (2022) set $g(r, p_0) = r$, which optimizes the log-likelihood of actions weighted by their observed rewards:

$$\hat{U}_n^{\text{LPI}}(\pi) = \frac{1}{n} \sum_{i=1}^n R_i \log \pi(A_i \mid X_i). \tag{21}$$

The asymptotic policy balances reward-seeking with imitation of the logging policy:

$$\pi_*^{\text{LPI}}(a \mid x) \propto r(x, a)\pi_0(a \mid x). \tag{22}$$

**Clipped LPI (`cLPI`).** uses importance-weight clipping, setting $g(r, p_0) = \frac{r}{\max(p_0, \tau)}$:

$$\hat{U}_n^{\text{cLPI}}(\pi) = \frac{1}{n} \sum_{i=1}^n \frac{R_i}{\max\{\pi_0(A_i \mid X_i), \tau\}} \log \pi(A_i \mid X_i). \tag{23}$$

In a similar spirit to `cIPS`, its asymptotic solution corrects for action frequency under $\pi_0$, down-weighting the influence of rare actions due to the clipping:

$$\pi_*^{\text{cLPI}}(a \mid x) \propto r(x, a)\frac{\pi_0(a \mid x)}{\max\{\pi_0(a \mid x), \tau\}}. \tag{24}$$

**KL regularization (`RegKL`).** To further amplify the reward signal relative to the logging policy prior, RegKL uses an exponential weighting function $g(r, p_0) = \exp(r/\beta)$:

$$\hat{U}_n^{\text{RegKL}}(\pi) = \frac{1}{n} \sum_{i=1}^{n} \exp(R_i/\beta) \log \pi(A_i \mid X_i). \tag{25}$$

The asymptotic policy is proportional to the logging policy, weighted by the exponentiated reward:

$$\pi_*^{\text{RegKL}}(a \mid x) \propto \mathbb{E}_{r \sim p(\cdot \mid x, a)}\big[\exp(r/\beta)\big]\pi_0(a \mid x). \tag{26}$$

The temperature parameter $\beta$ smoothly interpolates between behavior cloning ($\beta \to \infty$) and greedy reward maximization ($\beta \to 0$).

Note that BPR (Rendle et al., 2012) can be seen as an approximate PWLL objective, and we included it in our experiments. In fact, this general form of PWLL lends itself to numerous variations by modifying the weighting function $g$. For instance, one could introduce variants inspired by regularized IPS like Exponential Smoothing (esLPI) to fine-tune the bias-variance trade-off. While many such variants can be proposed for specific use cases, the central message of our work is that the well-behaved optimization landscape of the PWLL family is of greater practical importance than the estimation accuracy of OPE objectives. Thus, an exploration of these PWLL variants is beyond our scope. We contend that the foundational methods analyzed above, LPI, cLPI, and RegKL, along with the widely used BPR are sufficient to demonstrate the inherent advantages of PWLL objectives.

Finally, PWLL resembles reward- or advantage-weighted behavioral cloning objectives in RL (Nair et al., 2020; Wang et al., 2020; Peng et al., 2019; Peters, 2006). While those methods address multi-step MDPs and often focus on mitigating distributional shift and bootstrapping errors, we focus on offline contextual bandits with large action spaces: identifying objectives and parametrizations that remain optimizable as $K$ grows, rather than accurately estimating $V(\pi)$. PWLL is critic-free and uses logged rewards and propensities through a weighting function $g(R_i, \pi_0(A_i \mid X_i))$ that induces concave optimization landscapes for common policy classes. This yields substantial gains in large-$K$ bandits without the overhead of value-function estimation. PWLL's optimization-centric perspective complements the usual KL-regularized or trust-region interpretations of these RL methods. However, extending this analysis rigorously to multi-step RL is left for future work.

## 4 EMPIRICAL ANALYSIS

We conduct our empirical evaluation on three large-scale recommendation datasets: MovieLens ($K = 60\text{k}$) (Lam & Herlocker, 2016), Twitch ($K = 200\text{k}$) (Rappaz et al., 2021), and GoodReads ($K = 1\text{M}$) (Wan et al., 2019). These benchmarks feature action spaces with up to one million items, representing some of the largest settings studied in the offline policy learning literature. For all experiments, we employ the common softmax inner-product policies. We compare methods from both objective families. For OPE objectives, we include IPS, ES, DR, MIPS, OffCEM, POTEC, and PC in Section 2. For PWLL objectives, we evaluate LPI, cLPI, RegKL, and BPR in Section 3. All implementation details are provided in Appendix F.

### 4.1 OPTIMIZATION IS THE MAIN BOTTLENECK

To test our central hypothesis that *optimization challenges are a more significant barrier than estimation accuracy*, we evaluate how OPL objectives perform under various optimization configurations. If an algorithm's success is highly dependent on specific hyperparameters like batch size or learning rate, it suggests a difficult, non-robust optimization landscape. This experiment directly probes the practical trainability of each method, a key aspect our paper argues is often overlooked.

The results strongly support our claim. As shown in Fig. 2, *OPE objectives are highly sensitive* to batch size and learning rate schedule: minor changes can cause performance collapse, making them difficult to tune and train reliably. In contrast, *PWLL objectives remain robust*, achieving consistently high reward across all configurations. This stability translates directly into better learned policies: *PWLL objectives outperform OPE objectives on all datasets*. Even POTEC, a state-of-the-art method designed for large action spaces, is surpassed by the much simpler and easier-to-optimize cLPI. This supports our central claim: *optimization stability is key to effective OPL*.

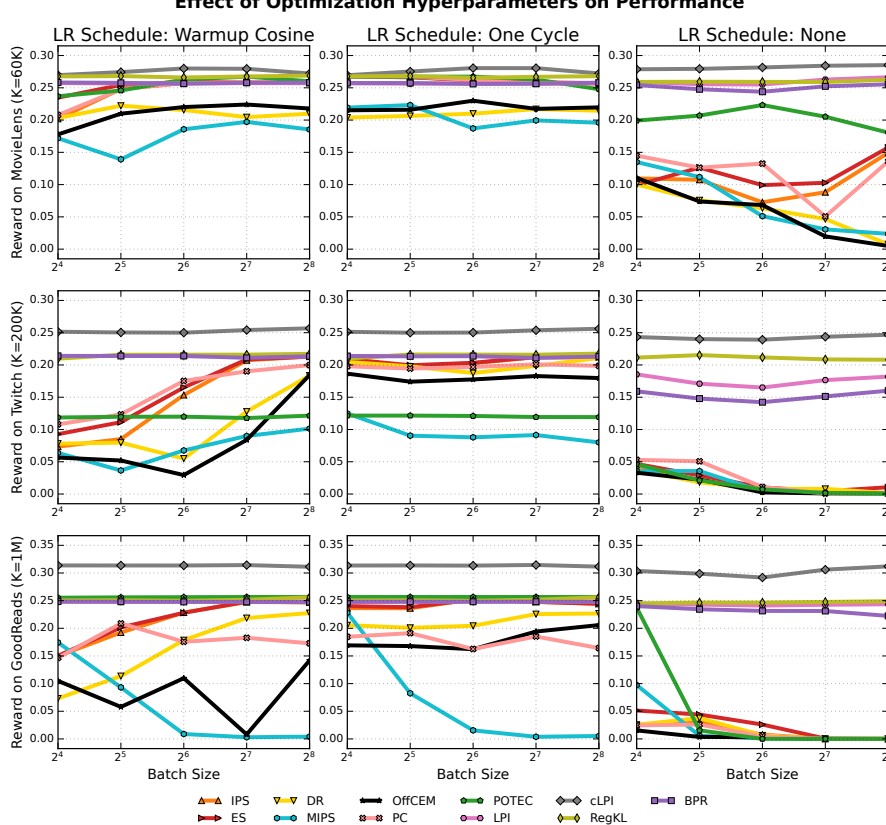

Figure 2: Effect of batch size and learning rate schedule on final validation reward using three large-scale datasets. OPE objectives are highly sensitive, while PWLL objectives are robust.

One might assume that an objective designed for estimation fidelity, such as a low-MSE OPE estimator, would naturally yield a better policy. Our findings show this is not the case. The superiority of PWLL objectives, which are poor value estimators by design, provides compelling evidence against this estimator-centric view. This reinforces our main takeaway: in large-scale OPL, a tractable optimization landscape is a more critical feature for a learning objective than its statistical accuracy. For completeness, an experiment tracking the MSE of methods is given in Appendix F.

The figure also supports Claim 2.4. Indeed, there is a consistent performance gap between POTEC and OffCEM. Both methods are designed to maximize the same asymptotic objective as we show in Section 2; their statistical goals are identical. The divergence in performance, therefore, can be attributed entirely to their differing optimization strategies. POTEC's use of a two-stage, cluster-level optimization proves far more effective than OffCEM's naive, action-level parametrization.

## 4.2 OBJECTIVE-AWARE PARAMETRIZATION

To empirically validate Claim 2.3, we compare a naive, whole-action-space parametrization against our proposed objective-aware approach, which restricts the policy's effective action space to the logging policy support, $S_0(x)$. As shown for the IPS objective in Fig. 3, the naive approach is highly unstable, with performance collapsing under simple learning configurations. In contrast, the objective-aware version is very robust, achieving high reward consistently across all batch sizes and schedules. This benefit extends even to inherently stable PWLL objectives like cLPI, which achieve even better performance with the restricted support. This provides strong evidence for Claim 2.3: aligning the policy structure with the objective's inductive bias simplifies the optimization landscape, leading to greater stability and superior learned policies. This finding holds across all datasets, with full results available in Appendix F.

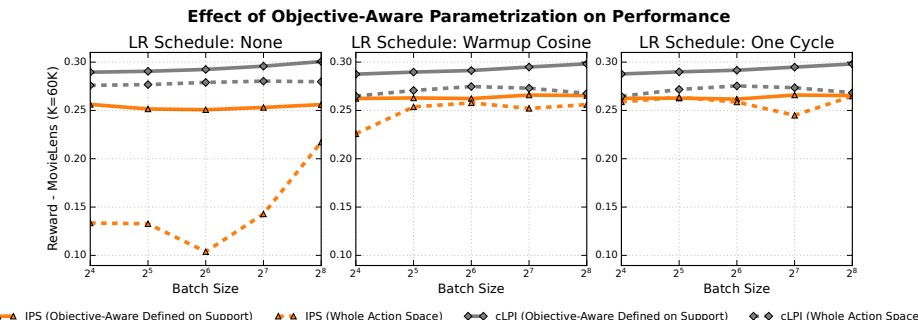

Figure 3: The effect of objective-aware parametrization for `IPS` and `cLPI` on `MovieLens`.

### 4.3 LIGHTWEIGHT POLICY PARAMETRIZATION HELPS

To further isolate the impact of policy complexity on optimization, we compare lightweight and heavyweight policy parametrizations (defined in Eq. (19)) for each objective. As shown for the `MovieLens` dataset in Fig. 4, the benefits of a simpler model are clear: lightweight policies converge faster and often achieve slightly higher final rewards. This finding reinforces a key theme of our work: a more tractable optimization landscape can be more critical to achieving strong performance than the greater expressive capacity of a complex policy. Results for other datasets are deferred to Appendix F.

Figure 4: Training progress over 10 epochs on `MovieLens`, comparing heavyweight vs. lightweight policies.

## 5 CONCLUSION

The common approach to OPL, which focuses on optimizing increasingly sophisticated OPE estimators, neglects a crucial factor: the optimization landscape. We show, both theoretically and empirically, that for large action spaces, this landscape becomes challenging, affecting the practical effectiveness of these methods. Our study of this landscape motivates clever policy parameterizations and PWLL objectives. By exploiting the estimator's inductive biases, these approaches alleviate optimization challenges and induce strong concavity for common policy classes. Our experiments confirm that this focus on optimization pays off: these simple changes make learning more robust, easier to tune, and converge to superior policies. This work advocates for a shift in focus for OPL research in large-scale settings, from estimator design towards the development of objectives with favorable optimization properties.

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

## A    EXTENDED RELATED WORK

**Offline contextual bandits.**    The contextual bandit framework is widely used for online learning under uncertainty (Lattimore & Szepesvari, 2019). Yet, many applications pose challenges for online exploration, motivating offline approaches that optimize decisions from logged data (Bottou et al., 2013). Since large datasets of past interactions are often available, policies can be improved without new experimentation (Swaminathan & Joachims, 2015a). This setting, known as offline (or off-policy) contextual bandits (Dudík et al., 2011), relies on off-policy evaluation (OPE) to estimate policy performance from logged data. These estimators are then used to learn value-maximizing policies (Off-policy learning, OPL).

**Off-policy evaluation.**    OPE (Dudík et al., 2011; Dudík et al., 2012; Dudik et al., 2014; Wang et al., 2017; Farajtabar et al., 2018; Su et al., 2020; Metelli et al., 2021; Kuzborskij et al., 2021; Saito & Joachims, 2022; Sakhi et al., 2020; Jeunen & Goethals, 2021) has attracted significant attention in recent years, with methods falling into three main categories. The direct method (DM) fits a model to predict expected costs for each context–action pair and then uses it to estimate policy value (Jeunen & Goethals, 2021; Aouali et al., 2025), a strategy that has proven particularly effective in large-scale recommender systems (Sakhi et al., 2020; Jeunen & Goethals, 2021). Inverse propensity scoring (IPS) instead reweights observed outcomes to correct for the bias of the logging policy (Horvitz & Thompson, 1952; Dudík et al., 2012). While IPS is unbiased under absolute continuity, it is highly sensitive to variance and bias when this condition is violated (Sachdeva et al., 2020). A wide range of techniques has been proposed to address this issue, including clipping (Ionides, 2008; Bottou et al., 2013), shrinkage (Su et al., 2020), smoothing (Metelli et al., 2021; Aouali et al., 2023; Sakhi et al., 2024), implicit exploration (Gabbianelli et al., 2024), and self-normalization (Swaminathan & Joachims, 2015b), among others (Aouali et al., 2024). A third line of work combines these two approaches in doubly robust (DR) estimators, which integrate modeling with reweighting for improved bias–variance trade-offs (Robins & Rotnitzky, 1995; Dudík et al., 2011; Dudik et al., 2014; Farajtabar et al., 2018). Our work focuses on off-policy learning using these estimators.

**Off-policy learning.**    OPL is typically built on DM, IPS, or DR. DM selects actions by maximizing predicted reward, either deterministically or stochastically, while IPS and DR optimize a parameterized policy via stochastic gradient descent (Swaminathan & Joachims, 2015a), where the unknown gradient of the true risk must be estimated using reweighting. Beyond these approaches, statistical learning tools have introduced new objectives grounded in PAC-based pessimism, providing stronger theoretical guarantees (London & Sandler, 2019; Sakhi et al., 2023a). Our contribution complements this literature by examining the optimization landscape of OPL in large action spaces, which remains largely underexplored.

**Large action spaces.**    Regularization can improve IPS in moderate settings, but scaling to large action spaces requires additional structure. One prominent direction leverages action embeddings: for example, marginalized IPS (MIPS) (Saito & Joachims, 2022) reduces variance by exploiting embedding information while remaining unbiased if the embeddings capture the causal effects of actions on costs. High-dimensional embeddings, however, can still induce variance, and misspecified embeddings can introduce bias. Recent work addresses these issues by learning embeddings directly from data (Peng et al., 2023; Cief et al., 2024) or relaxing causal assumptions (Taufiq et al., 2024; Saito et al., 2023). A complementary line of research addresses computational challenges: training policies over large action spaces scales linearly with the number of actions $K$, motivating fast maximum inner product search (MIPS) techniques (Shrivastava & Li, 2014; Sakhi et al., 2023c) to reduce complexity.

Beyond the offline setting, several works study large action spaces in the *online* contextual bandit framework (Foster et al., 2020; Xu & Zeevi, 2020; Zhu et al., 2022; Aouali, 2025). These approaches rely on active exploration and repeated interaction with the environment, enabling algorithms to gather information adaptively. Their assumptions and techniques are therefore not applicable to the offline regime we consider, where learning must rely entirely on a fixed, biased log of historical actions. Unlike this online literature, our work investigates the optimization landscape specific to offline learning in large action spaces and provides practical and theoretical insights on how to make these offline objectives more amenable to gradient-based optimization. We view this as a fundamental yet relatively unexplored research direction.

## B    ADDITIONAL OBJECTIVES AND THEIR SSYMPTOTIC SOLUTIONS

All these previous estimators are linear on the policy $\pi$. Another way of reducing variance is smoothing the importance weights, leading to non-linear estimators on the policy and obtaining stochastic, non-deterministic optimal solutions.

**IW Exponential Smoothing (IW-ES)**    Exponential Smoothing (Aouali et al., 2023) can also be applied on the Importance weights themselves instead of the logging propensity, resulting in an estimator:

$$\hat{V}_n^{\text{IW-ES}}(\pi) = \frac{1}{n} \sum_{i=1}^{n} \left( \frac{\pi(A_i|X_i)}{\pi_0(A_i|X_i)} \right)^{\alpha} R_i \,.$$

This estimator recovers the following when $n \to \infty$:

$$V^{\text{IW-ES}}(\pi) = \mathbb{E}_{x \sim \nu, a \sim \pi_0(\cdot|x)} \left[ \left( \frac{\pi(a|x)}{\pi_0(a|x)} \right)^{\alpha} r(x,a) \right]$$

$$= \mathbb{E}_{x \sim \nu, a \sim \pi(\cdot|x)} \left[ \left( \frac{\pi(a|x)}{\pi_0(a|x)} \right)^{\alpha-1} r(x,a) \right] \,.$$

To identify the optimal policy, we need to find the maximizer of this objective w.r.t $\pi$. This objective decomposes on the contexts $x$, meaning that its global minimizer is a minimizer for each $x$ dependent sub-problem. For each $x \in \mathcal{X}$, we can write down our maximization objective as:

$$\max_{\pi(\cdot|x)} \mathbb{E}_{a \sim \pi(\cdot|x)} \left[ \left( \frac{\pi(a|x)}{\pi_0(a|x)} \right)^{\alpha-1} r(x,a) \right]$$

$$\text{s.t} \quad \sum_{a \in \mathcal{A}} \pi(a|x) = 1.$$

$$\forall a \in \mathcal{A}, \pi(a|x) \geq 0.$$

This can be solved by setting the Lagragian to $0$. There exists a $\lambda \geq 0$ with which the optimal policy $\pi_*^{\text{IW-ES}}$ verifies:

$$L_\lambda^{\text{IW-ES}}(\pi_*^{\text{IW-ES}}) = 0 \iff \alpha \left( \frac{\pi_*^{\text{IW-ES}}(a|x)}{\pi_0(a|x)} \right)^{\alpha-1} r(x,a) - \lambda = 0$$

$$\iff \pi_*^{\text{IW-ES}}(a|x) \propto r(x,a)^{1/(1-\alpha)} \pi_0(a|x) \,,$$

which ends the proof.

**Logarithmic Smoothing (LS)**    Hard clipping the weights makes the learning problem non-smooth, with an optimal solution hard to derive. We focus on a new estimator that has been proposed (Sakhi et al., 2024), which can be interpreted as soft clipping, with strong concentration guarantees that logarithmically smooths the importance weights of the estimator with $\lambda \geq 0$:

$$\hat{V}_n^{\text{LS-}\lambda}(\pi) = \frac{1}{n} \sum_{i=1}^{n} \frac{1}{\lambda} \log \left( 1 + \lambda \frac{\pi(A_i|X_i)}{\pi_0(A_i|X_i)} \right) R_i \,. \tag{27}$$

Note that the LS estimator was introduced with the reward inside the log but we stick to this definition for the ease of derivations it brings. We can derive its optimal policy by solving the following for any $x \in \mathcal{X}$:

$$\pi_*^{\text{LS-}\lambda}(a|x) \propto \frac{1}{\lambda} \left( \frac{r(x,a)}{C_0 + L_a} - 1 \right) \pi_0(a|x)$$

$$\lambda + 1 = \mathbb{E}_{\pi_0(\cdot|x)} \left[ \frac{r(x,a)}{C_0 + L_a} \right]$$

$$\forall a \in \mathcal{A}, \quad L_a = 0 \iff \pi(a|x) > 0$$

$$\iff r(x,a) > C + L_a \,,$$

with $C_0$ and $L_a$ positive slack variables. We can observe that the solution interpolates between smooth, full support solution and a degenerate policy (with 0 probability mass for some actions) depending on the value of $\lambda$. Precisely, $\lambda \to 0$ recovers the `IPS` solution (as `LS` converge towards `IPS`) while $\lambda$ big enough recovers a smoother solution, linear on the reward and proportional to:

$$\pi_*^{\texttt{LS}-\lambda}(a|x) \propto \frac{1}{\lambda}\left(\frac{(\lambda+1)r(x,a)}{\mathbb{E}_{\pi_0(\cdot|x)}\left[r(x,a)\right]} - 1\right)\pi_0(a|x)\mathbb{1}\left[r(x,a) > \mathbb{E}_{\pi_0(\cdot|x)}\left[r(x,a)\right]/(\lambda+1)\right].$$

This means that if we are interested in reaching a smooth solution, we need to increase $\lambda$.

## C    DERIVATIONS OF ASYMPTOTIC SOLUTIONS

### C.1    ASYMPTOTIC SOLUTIONS FOR OPE OBJECTIVES

**(IPS), its Clipping Variant (cIPS) and ES**    Recall the definition of the (logging propensity) clipped IPS estimator with $\tau \in [0, 1]$:

$$\hat{V}_n^{\texttt{cIPS}}(\pi) = \frac{1}{n}\sum_{i=1}^n \frac{\pi(A_i|X_i)}{\max\{\pi_0(A_i|X_i), \tau\}}R_i\,.$$

Taking $n \to \infty$, one obtains:

$$V^{\texttt{cIPS}}(\pi) = \mathbb{E}_{X\sim\nu, A\sim\pi_0(\cdot|x)}\left[\frac{\pi(a|x)}{\max\{\pi_0(a|x), \tau\}}r(x,a)\right]$$

$$= \mathbb{E}_{X\sim\nu, A\sim\pi(\cdot|x)}\left[\frac{\pi_0(a|x)}{\max\{\pi_0(a|x), \tau\}}r(x,a)\right].$$

As the objective is linear in the policy $\pi$, the optimal policy should put for any $x \in \mathcal{X}$, all the mass on the action $a$ that maximizes the weighted reward, giving:

$$\pi_*^{\texttt{cIPS}}(a|x) = \mathbb{1}\left[a = \operatorname*{argmax}_{a'\in\mathcal{A}} \frac{\pi_0(a'|x)r(x,a')}{\max\{\pi_0(a'|x), \tau\}}\right].$$

We recover the solution for `IPS` when we tend $\tau \to 0$:

$$\pi_*^{\texttt{IPS}}(a|x) = \mathbb{1}\left[a = \operatorname*{argmax}_{a'\in\mathcal{A}} r(x,a')\mathbb{1}\left[\pi_0(a'|x) > 0\right]\right].$$

We also recover the solution of `ES` just by replacing the clipping function by an exponential function of factor $\alpha$, obtaining:

$$\pi_*^{\texttt{ES}}(a|x) = \mathbb{1}\left[a = \operatorname*{argmax}_{a'\in\mathcal{A}} r(x,a')\pi_0(a'|x)^{1-\alpha}\right].$$

**Doubly Robust (DR)**    The doubly robust estimator converges to the following quantity:

$$V^{\texttt{DR}}(\pi) = \mathbb{E}_{X\sim\nu, A\sim\pi_0(\cdot|x)}\left[\frac{\pi(a|x)}{\max\{\pi_0(a|x), \tau\}}(r(x,a) - \hat{r}(x,a))\right] + \mathbb{E}_{x\sim\nu, a\sim\pi(\cdot|x)}\left[\hat{r}(x,a)\right]$$

$$= \mathbb{E}_{x\sim\nu, a\sim\pi(\cdot|x)}\left[(r(x,a) - \hat{r}(x,a))\frac{\pi_0(a|x)}{\max\{\pi_0(a|x), \tau\}} + \hat{r}(x,a)\right].$$

The objective is linear in $\pi$ and is thus maximized by the following deterministic decision rule:

$$\pi_*^{\texttt{DR}}(a|x) = \mathbb{1}\left[a = \operatorname*{argmax}_{a'\in\mathcal{A}} \hat{r}(x,a') + (r(x,a') - \hat{r}(x,a'))\frac{\pi_0(a|x)}{\max\{\pi_0(a|x), \tau\}}\right]$$

**Marginalized IPS (MIPS) with clusters.**    We adopt the same approach to look for the minimizer of `MIPS`. We generalize the clustering function $h$ to take also the context into account. We write down the estimator:

$$\hat{V}_n^{\texttt{MIPS}}(\pi) = \frac{1}{n}\sum_{i=1}^n \frac{\sum_{a'}\mathbb{1}\left[h(a', X_i) = h(A_i, X_i)\right]\pi(a'|X_i)}{\sum_{a''}\mathbb{1}\left[h(a'', X_i) = h(A_i, X_i)\right]\pi_0(a''|X_i)}R_i = \frac{1}{n}\sum_{i=1}^n \frac{\pi(C_i|X_i)}{\pi_0(C_i|X_i)}R_i\,,$$

with which, we recover when $n \to \infty$:

$$V^{\text{MIPS}}(\pi) = \mathbb{E}_{X \sim \nu, A \sim \pi_0(\cdot|x)} \left[ \frac{\sum_{a'} \mathbb{1}\left[h(a',x) = h(a,x)\right] \pi(a'|x)}{\sum_{a''} \mathbb{1}\left[h(a'',x) = h(a,x)\right] \pi_0(a''|x)} r(x,a) \right]$$

$$= \mathbb{E}_{x \sim \nu} \left[ \sum_a \pi_0(a|x) \frac{\sum_{a'} \mathbb{1}\left[h(a',x) = h(a,x)\right] \pi(a'|x)}{\sum_{a''} \mathbb{1}\left[h(a'',x) = h(a,x)\right] \pi_0(a''|x)} r(x,a) \right]$$

$$= \mathbb{E}_{x \sim \nu} \left[ \sum_{a'} \pi(a'|x) \sum_a \pi_0(a|x) \frac{\mathbb{1}\left[h(a',x) = h(a,x)\right]}{\sum_{a''} \mathbb{1}\left[h(a'',x) = h(a,x)\right] \pi_0(a''|x)} r(x,a) \right]$$

$$= \mathbb{E}_{x \sim \nu} \left[ \sum_{a'} \pi(a'|x) \mathbb{E}_{a \sim \pi_0(\cdot|x)} \left[ \frac{\mathbb{1}\left[h(a',x) = h(a,x)\right] r(x,a)}{\mathbb{E}_{a'' \sim \pi_0(\cdot|x)} \left[ \mathbb{1}\left[h(a'',x) = h(a,x)\right] \right]} \right] \right].$$

The objective is linear in $\pi$, and depends on the action $a'$ through its cluster $h(a, \cdot)$ alone. This means that multiple solutions are maximizers as long as the policy chooses the best cluster $c$. We thus write down the asymptotic solution for $\text{MIPS}$ in the cluster level, giving:

$$\pi_*^{\text{MIPS}}(c|x) = \mathbb{1}\left[ c = \underset{c' \in \mathcal{C}}{\arg\max} \left\{ \mathbb{E}_{a \sim \pi_0(\cdot|x)} \left[ \frac{r(x,a) \mathbb{1}[h(a,x) = c']}{\mathbb{E}_{a'' \sim \pi_0(\cdot|x)} \left[ \mathbb{1}\left[h(a'',x) = h(a,x)\right] \right]} \right] \right\} \right]$$

$$= \mathbb{1}\left[ c = \underset{c' \in \mathcal{C}}{\arg\max} \left\{ \mathbb{E}_{a \sim \pi_0(\cdot|x)} \left[ \frac{r(x,a) \mathbb{1}[h(a,x) = c']}{\mathbb{E}_{a'' \sim \pi_0(\cdot|x)} \left[ \mathbb{1}\left[h(a'',x) = c'\right] \right]} \right] \right\} \right]$$

$$= \mathbb{1}\left[ c = \underset{c' \in \mathcal{C}}{\arg\max} \left\{ \frac{\mathbb{E}_{a \sim \pi_0(\cdot|x)} \left[ r(x,a) \mathbb{1}[h(a,x) = c'] \right]}{\mathbb{E}_{a \sim \pi_0(\cdot|x)} \left[ \mathbb{1}\left[h(a,x) = c'\right] \right]} \right\} \right],$$

which ends the proof.

**Conjunct Effect Modeling (OffCEM).** This estimator can be seen as the natural, doubly robust extension of the $\text{MIPS}$ estimator.

$$\hat{V}_n^{\text{OffCEM}}(\pi) = \frac{1}{n} \sum_{i=1}^n \frac{\pi(C_i|X_i)}{\pi_0(C_i|X_i)} \left( R_i - \hat{r}(A_i, X_i) \right) + \mathbb{E}_{a \sim \pi(\cdot|X_i)} \left[ \hat{r}(X_i, a) \right].$$

We take the sample size $n$ to $\infty$ and obtain:

$$V^{\text{OffCEM}}(\pi) = \mathbb{E}_{X \sim \nu, A \sim \pi_0(\cdot|x)} \left[ \frac{\sum_{a'} \mathbb{1}\left[h(a',x) = h(a,x)\right] \pi(a'|x)}{\sum_{a''} \mathbb{1}\left[h(a'',x) = h(a,x)\right] \pi_0(a''|x)} (r(x,a) - \hat{r}(x,a)) \right] + \mathbb{E}_{x \sim \nu, a \sim \pi(\cdot|x)} \left[ \hat{r}(x,a) \right]$$

$$= \mathbb{E}_{x \sim \nu, a' \sim \pi(\cdot|x)} \left[ \mathbb{E}_{a \sim \pi_0(\cdot|x)} \left[ \frac{\mathbb{1}\left[h(a',x) = h(a,x)\right] (r(x,a') - \hat{r}(x,a'))}{\mathbb{E}_{a'' \sim \pi_0(\cdot|x)} \left[ \mathbb{1}\left[h(a'',x) = h(a,x)\right] \right]} \right] \right] + \mathbb{E}_{x \sim \nu, a' \sim \pi(\cdot|x)} \left[ \hat{r}(x,a') \right]$$

$$= \mathbb{E}_{x \sim \nu, a' \sim \pi(\cdot|x)} \left[ \mathbb{E}_{a \sim \pi_0(\cdot|x)} \left[ \frac{\mathbb{1}\left[h(a',x) = h(a,x)\right] (r(x,a') - \hat{r}(x,a'))}{\mathbb{E}_{a'' \sim \pi_0(\cdot|x)} \left[ \mathbb{1}\left[h(a'',x) = h(a,x)\right] \right]} \right] + \hat{r}(x,a') \right].$$

The solution depends explicitly on the action $a$ by the reward model $\hat{r}$. We can derive it as the objective is linear on $\pi$, obtaining:

$$\pi_*^{\text{OffCEM}}(a|x) = \mathbb{1}\left[ a = \underset{a' \in \mathcal{A}}{\arg\max} \left\{ \hat{r}(x,a') + \frac{\mathbb{E}_{\bar{a} \sim \pi_0(\cdot|x)} \left[ r(\bar{a},x) - \hat{r}(\bar{a},x) \mathbb{1}[h(x,\bar{a}) = h(x,a')] \right]}{\pi_0(h(x,a')|x)} \right\} \right].$$

**Two Stage Decomposition (POTEC).** This is an *optimization strategy* for $\text{OffCEM}$. It restricts the policy to a cluster-informed form,

$$\pi(a \mid x) = \sum_{c \in \mathcal{C}} \pi^{\text{RM}}(a \mid x, c) \pi^{\text{CL}}(c \mid x),$$

where $\pi^{\text{RM}}(a \mid x, c) = \mathbb{1}[a = \arg\max_{a' \in c} \hat{r}(x,a')]$ is fixed, model-based policy that deterministically selects the best action within each cluster. Learning is then simplified to finding the optimal cluster-level policy $\pi^{\text{CL}}$ that maximizes the $\text{OffCEM}$ objective in :

$$\hat{V}_n^{\text{POTEC}}(\pi^{\text{CL}}) = \frac{1}{n} \sum_{i=1}^n \left( \frac{\pi^{\text{CL}}(C_i \mid X_i)}{\pi_0(C_i \mid X_i)} \left( R_i - \hat{r}(X_i, A_i) \right) + \sum_{c \in \mathcal{C}} \pi^{\text{CL}}(c \mid X_i) \hat{r}_c^*(X_i) \right),$$

This is exactly the Doubly Robust version of MIPS on the cluster level, the asymptotic solution on the cluster level can be followed in the same fashion:

$$\pi_*^{\text{CL}}(c \mid x) = \mathbb{1}\left[c = \operatorname*{argmax}_{c' \in \mathcal{C}} \left\{ \frac{\mathbb{E}_{a \sim \pi_0(\cdot | x)}\left[(r(x,a) - \hat{r}(x,a))\mathbb{1}[h(a,x) = c']\right]}{\mathbb{E}_{a \sim \pi_0(\cdot | x)}\left[\mathbb{1}\left[h(a,x) = c'\right]\right]} + \hat{r}_{c'}^*(x)\right\}\right].$$

The optimal policy for the POTEC optimization strategy unfolds as:

$$\pi_*^{\text{POTEC}}(a | x) = \sum_{c \in \mathcal{C}} \pi^{\text{RM}}(a \mid x, c)\pi_*^{\text{CL}}(c \mid x).$$

At first glance, it might be hard to see the connection between POTEC and OffCEM solutions, but they are equivalent. For ease of notation, let us denote by $D_{\hat{r},x}(c)$:

$$D_{\hat{r},x}(c) = \frac{\mathbb{E}_{a \sim \pi_0(\cdot | x)}\left[(r(x,a) - \hat{r}(x,a))\mathbb{1}[h(a,x) = c]\right]}{\mathbb{E}_{a \sim \pi_0(\cdot | x)}\left[\mathbb{1}\left[h(a,x) = c\right]\right]}.$$

and recall that the optimal policy of OffCEM finds the action $a$ that maximizes:

$$\tilde{V}(x,a) = \hat{r}(x,a) + D_{\hat{r},x}(h(a,x)).$$

For any context $x$, the optimal action $a^*$ of POTEC verifies:

- $a^*$ is in the optimal cluster: $h(a^*,x) = c_*(x)$ with $c_*(x) = \operatorname{argmax}_{c \in \mathcal{C}} D_{\hat{r},x}(c) + \hat{r}_c^*(x)$.
- $a^*$ is optimal within that cluster: $a = \operatorname{argmax}_{a \in c_*(x)} \hat{r}(x,a)$.

This means that for all actions $a$ with $h(a,x) \neq c_*(x)$, we have:

$$\begin{aligned}
\tilde{V}(x,a) &= D_{\hat{r},x}(h(a,x)) + \hat{r}(x,a)\\
&\leq D_{\hat{r},x}(h(a,x)) + \hat{r}_{h(a,x)}^*(x)\\
&\leq D_{\hat{r},x}(c_*(x)) + \hat{r}_{c_*(x)}^*(x)\\
&= D_{\hat{r},x}(h(x,a^*)) + \hat{r}(x,a^*) = \tilde{V}(x,a^*).
\end{aligned}$$

In addition, for all actions $a$ with $h(a,x) = c_*(x)$, we have:

$$\begin{aligned}
\tilde{V}(x,a) &= D_{\hat{r},x}(h(a,x)) + \hat{r}(x,a)\\
&= D_{\hat{r},x}(c_*(x)) + \hat{r}(x,a)\\
&\leq D_{\hat{r},x}(c_*(x)) + \hat{r}_{c_*(x)}^*(x) = \tilde{V}(x,a^*).
\end{aligned}$$

This means that the optimal action $a^*$ for POTEC is the maximizer of $\tilde{V}(x,a)$, which is exactly the solution of OffCEM.

**Policy Convolution (PC)** This estimator uses a nearest neighbors function to aggregate the propensities of similar actions, making the hypothesis that similar actions will result in similar reward signal. The esimator writes:

$$\hat{V}_n^{\text{PC}}(\pi) = \frac{1}{n}\sum_{i=1}^n \frac{\pi(N_\epsilon(A_i) \mid X_i)}{\pi_0(N_\epsilon(A_i) \mid X_i)} R_i, \quad \text{with } \pi(N_\epsilon(a) \mid x) = \sum_{a' \in N_\epsilon(a)} \pi(a' \mid x).$$

This estimator is equivalent to the following when $n \to \infty$:

$$\begin{aligned}
V^{\text{PC}}(\pi) &= \mathbb{E}_{X \sim \nu, A \sim \pi_0(\cdot | x)}\left[\frac{\sum_{a'} \pi(a' | x)\mathbb{1}\left[a' \in N_\epsilon(a)\right]}{\pi_0(N_\epsilon(a) | x)} r(X,A)\right]\\
&= \mathbb{E}_{x \sim \nu, a \sim \pi(\cdot | x)}\left[\mathbb{E}_{\bar{a} \sim \pi_0(\cdot | x)}\left[\frac{r(\bar{a},x)\mathbb{1}\left[a \in N_\epsilon(\bar{a})\right]}{\pi_0(N_\epsilon(\bar{a}) | x)}\right]\right].
\end{aligned}$$

The same argument of linearity applies here, giving us the corresponding asymptotic solution:

$$\pi_*^{\text{PC}}(a | x) = \mathbb{1}\left[a = \operatorname*{argmax}_{a' \in \mathcal{A}} \left\{\mathbb{E}_{\bar{a} \sim \pi_0(\cdot | x)}\left[\frac{r(x,\bar{a})\mathbb{1}[a' \in N_\epsilon(\bar{a})]}{\pi_0(N_\epsilon(\bar{a}) | x)}\right]\right\}\right].$$

### C.2 Asymptotic solutions for PWLL objectives

Our objectives can be written in the same form, only choosing for each a different function $g$:

$$\hat{U}_n^g(\pi) = \frac{1}{n} \sum_{i=1}^n g(X_i, A_i, R_i) \log \pi(A_i \mid X_i).$$

We solve the maximization of the asymptotic value of this objective, to recover the solutions of our surrogate objectives as a special case. The objective decomposes on the contexts $x$, we are thus interested in the following maximization problem for each $x$:

$$\max_{\pi(\cdot|x)} \mathbb{E}_{a \sim \pi_0(\cdot|x)} \left[ \mathbb{E}_r \left[ g(x, a, r) \right] \log \pi(a|x) \right]$$

$$\text{s.t} \quad \sum_{a \in \mathcal{A}} \pi(a|x) = 1.$$

$$\forall a \in \mathcal{A}, \pi(a|x) \geq 0.$$

This can be solved by setting the Lagrangian to $0$. There exists a $\lambda \geq 0$ with which the optimal policy $\pi_*^g$ verifies for all $x$ and $a$:

$$L_\lambda^g(\pi_*^g)(x, a) = 0 \iff \pi_0(a|x) \frac{1}{\pi_*^g(a|x)} \mathbb{E}_r \left[ g(x, a, r) \right] - \lambda = 0$$

$$\iff \pi_*^g(a|x) \propto \mathbb{E}_r \left[ g(x, a, r) \right] \pi_0(a|x),$$

which concludes the proof.

## D Proofs for optimization properties

In this section, we prove the propositions about the optimization landscape of OPE based and PWLL based learning approaches. We start by stating the following lemmas, that will be helpful to prove our propositions.

**Lemma D.1.** *For all* $\mathbf{r} \in [0, 1]^K$, $\theta \mapsto \hat{V}_n(\pi_\theta)$ *is* $5/2$-*smooth, i.e., for all* $\pi_\theta := softmax(\theta)$ *and* $\pi_{\theta'} := softmax(\theta')$, *we have,*

$$\left| \hat{V}_n(\pi_{\theta'}) - \hat{V}_n(\pi_\theta) - \left\langle \frac{d\hat{V}_n(\pi_\theta)}{d\theta}, \theta' - \theta \right\rangle \right| \leq \frac{5}{4} \cdot \|\theta' - \theta\|_2^2.$$

*Proof.* See the proof in (Mei et al., 2020b, Lemma 2). $\qquad\square$

**Lemma D.2.** *All the action level estimators* EST *in (*IPS, cIPS, DR, PC*) can be written, for any policy* $\pi$, *in the form:*

$$\hat{V}_n^{EST}(\pi) = \frac{1}{n} \sum_{i=1}^n \mathbb{E}_{a \sim \pi(\cdot|X_i)} \left[ \hat{r}_{EST,i}(a, X_i) \right], \tag{28}$$

*For the cluster level estimators/approaches* EST-C *in (*MIPS, POTEC*), we also have*

$$\hat{V}_n^{EST-C}(\pi) = \frac{1}{n} \sum_{i=1}^n \mathbb{E}_{c \sim \pi(\cdot|X_i)} \left[ \hat{r}_{EST-C,i}(c, X_i) \right], \tag{29}$$

*meaning that all these estimators are linear in* $\pi$.

*Proof.* This is straightforward to prove. We begin by the action level estimators and take DR as a representative. For DR, we have the following:

$$\hat{r}_{DR,i}(a, X_i) = \hat{r}(a, X_i) + \mathbb{I}[a = A_i] \frac{R_i - \hat{r}(A_i, X_i)}{\max(\tau, \pi_0(A_i|X_i))}$$

verifies the equation. Solutions for `cIPS` and `IPS` can be recovered directly, and `PC` follows the same construction. For the cluster level approaches, we take `POTEC` as a representative, and we have:

$$\hat{r}_{\text{POTEC},i}(c, X_i) = \hat{r}_c^\star(X_i) + \mathbb{I}[c = X_i]\frac{R_i - \hat{r}(A_i, X_i)}{\pi_0(C_i|X_i)}\,,$$

The $\hat{r}_{\text{MIPS},i}$ follows as a special case when $\hat{r} = 0$. $\qquad\qquad\square$

**Lemma D.3.** *For problems with a single state $x$, and for any EST in (`IPS`, `cIPS`, `DR`, `OffCEM`, `MIPS`, `PC`), there is a problem, defined by $r, \pi_0$ for which we obtain:*

$$\hat{r}_{EST}(a) = \mathbb{1}\left[a = a_K\right]\,, \tag{30}$$

*with $a_K$ being the optimal action. We can also find a problem, for any cluster level approach (`POTEC` and also `MIPS`) with:*

$$\hat{r}_{EST\text{-}C}(c) = \mathbb{1}\left[c = c_{|\mathcal{C}|}\right]\,, \tag{31}$$

*with $c_{|\mathcal{C}|}$ being the optimal cluster.*

*Proof.* Let us prove this for the `cIPS` for the action level estimators and `POTEC` for the cluster level approach. The result can be adapted for other estimators as they follow the same construction. To simplify the analysis, we suppose that $\pi_0(a) > 0$ for all actions. For `cIPS`, we have for any $\tau \in [0, 1]$, for $n$ large enough:

$$\hat{r}_{\text{EST}}(a) = \frac{\pi_0(a)}{\max(\pi_0(a), \tau)}r(a)\,.$$

We consider the problem where actual rewards are of the following form:

$$r(a) = \mathbb{1}\left[a = a_K\right]\frac{\max(\pi_0(a), \tau)}{\pi_0(a)} \in \mathbb{R}^+,$$

However, as the rewards need to be in $[0, 1]$, we choose $\tau < \max_a \pi(a)$. We can choose $a_K = \operatorname{argmax}_a \frac{\pi_0(a)}{\max(\pi_0(a), \tau)}$, and obtain $r(a_K) = 1$ and $r(a) = 0$ otherwise, giving $r \in [0, 1]$, and proving the existence of the problem. The same construction follows for `IPS`, `ES` and `DR`.

For the `POTEC` approach, we get also, for $n$ large enough:

$$\hat{r}_{\text{POTEC}}(c) = \max_{a \in c}\hat{r}(a) + \frac{\sum_{a \in c}\pi_0(a)(r(a) - \hat{r}(a))}{\pi_0(c)}$$

We define the problem where action $a_K$, the optimal action is really far from all the other actions in the cluster space, to get the action $a_K$ with its own cluster, i.e. $c_{|\mathcal{C}|} = h(a_K) = \{a_K\}$, and $\forall a \neq a_K, a_K \notin h(a)$. If we choose $r(a_K) = 1$ and $r(a) = 0$ otherwise, and additionally $\hat{r}(a_K) = 1 - \epsilon$ and $\hat{r}(a) = \epsilon$ otherwise, we get:

$$\hat{r}_{\text{POTEC}}(c) = \mathbb{1}\left[c = c_{|\mathcal{C}|}\right]\,.$$

This proves that there is a configuration in which the estimator has a one-hot reward. The same constructions can be done for all other estimators, based on the same ideas. $\qquad\square$

Now we restate Proposition 2.1 and proceed to its proof.

**Proposition D.4.** *Let $\hat{V}_n$ an OPE estimator linear in $\pi$, $\pi_n$ its maximizer. Let $\eta \in (0, 1]$ a learning rate. Even for a single context $x$, and a linear softmax policy $\pi_\theta(a) = \exp(\theta_a)/\sum_{a' \in \mathcal{A}_{\text{EFF}}}\exp(\theta_{a'})$ with $\mathcal{A}_{\text{EFF}}$ its effective action space, there exist a problem such that gradient descent **cannot escape a suboptimal region** before $t_0 = C.K_{\text{EFF}} = \mathcal{O}(K_{\text{EFF}})$ as we have:*

$$\forall t \leq t_0 : \hat{V}_n(\pi_n) - \hat{V}_n(\pi_{\theta_t}) \geq 0.9$$

*Proof.* The proof of this result follows the same technique as (Mei et al., 2020a, Theorem 1). We adapt it here and derive it for the sake of completeness. Let EST be one of the estimators with action level policies considered before and let $\hat{V}_n$ be that estimator. Consider the single context case where:

$$\hat{r}_{\text{EST}}(a) = \mathbb{1}\left[a = a_K\right].\tag{32}$$

This case exists per Lemma D.3. This means that for a policy $\pi$:

$$\hat{V}_n(\pi) = \pi(a_K).$$

If $a_K \in \mathcal{A}_{\text{EFF}}$, this means that the maximizer $\pi_n = \text{argmax}_{\pi_\theta} \hat{V}_n(\pi_\theta)$, reaches $\hat{V}_n(\pi_n) = 1$ and we have for any $\pi_\theta$:

$$r(a_K) - \hat{V}_n(\pi_\theta) = 1 - \pi_\theta(a_K).$$
$$r(a) - \hat{V}_n(\pi_\theta) = -\pi_\theta(a_K), \forall a \neq a_K.$$

The condition $a_K \in \mathcal{A}_{\text{EFF}}$ is important, as it ensures us that the considered family of parametrized policies include the optimal policy. This is the reason why $\mathcal{A}_{\text{EFF}}$ should be constructed using information from the optimal policy for the estimator optimized. For our policy, parametrized by a softmax $\pi_\theta(a) \propto \exp(\theta_a)\mathbb{I}[a \in \mathcal{A}_{\text{EFF}}]$, the $\ell_2$ norm of the gradient is upper bounded by:

$$\left\|\frac{d\hat{V}_n(\pi_\theta)}{d\theta}\right\|_2 = \sqrt{\pi_\theta(a_K)^2(1 - \pi_\theta(a_K))^2 + \pi_\theta(a_K)^2 \sum_{a \neq K} \pi_\theta(a)^2}$$

$$= \pi_\theta(a_K) \cdot \sqrt{(1 - \pi_\theta(a_K))^2 + \sum_{a \neq K} \pi_\theta(a)^2}$$

$$\leq \pi_\theta(a_K) \cdot \sqrt{(1 - \pi_\theta(a_K))^2 + \left(\sum_{a \neq K} \pi_\theta(a)\right)^2}$$

$$= \pi_\theta(a_K) \cdot \sqrt{(1 - \pi_\theta(a_K))^2 + (1 - \pi_\theta(a_K))^2}$$

$$= \sqrt{2} \cdot \pi_\theta(a_K) \cdot (1 - \pi_\theta(a_K)).$$

Let $\theta_{t+1} \leftarrow \theta_t + \eta_t \cdot \frac{d\hat{V}_n(\pi_{\theta_t})}{d\theta_t}$, and $\pi_{\theta_{t+1}} = \text{softmax}(\theta_{t+1})$ be the next policy after one step gradient update. Define the following two kinds of iterations:

$$t_{\text{good}} := \left\{t \geq 1 : \pi_{\theta_{t+1}}(a_K) > \pi_{\theta_t}(a_K)\right\},$$

$$t_{\text{bad}} := \left\{t \geq 1 : \pi_{\theta_{t+1}}(a_K) \leq \pi_{\theta_t}(a_K)\right\}.$$

For all $t \in t_{\text{bad}}$, we have,

$$\frac{1}{\pi_{\theta_t}(a_K)} - \frac{1}{\pi_{\theta_{t+1}}(a_K)} = \frac{1}{\pi_{\theta_{t+1}}(a_K) \cdot \pi_{\theta_t}(a_K)} \cdot \left(\pi_{\theta_{t+1}}(a_K) - \pi_{\theta_t}(a_K)\right) \leq 0.$$

For all $t \in t_{\text{good}}$, we have,

$$
\pi_{\theta_{t+1}}(a_K) - \pi_{\theta_t}(a_K) = \left[ \hat{V}_n(\pi_{\theta_{t+1}}) - \hat{V}_n(\pi_{\theta_t}) \right]
$$

$$
= \left[ \hat{V}_n(\pi_{\theta_{t+1}}) - \hat{V}_n(\pi_{\theta_t}) - \left\langle \frac{d\hat{V}_n(\pi_{\theta_t})}{d\theta_t}, \theta_{t+1} - \theta_t \right\rangle + \left\langle \frac{d\hat{V}_n(\pi_{\theta_t})}{d\theta_t}, \theta_{t+1} - \theta_t \right\rangle \right]
$$

$$
\leq \left[ \frac{5}{4} \cdot \|\theta_{t+1} - \theta_t\|_2^2 + \left\langle \frac{d\hat{V}_n(\pi_{\theta_t})}{d\theta_t}, \theta_{t+1} - \theta_t \right\rangle \right] \quad \text{(by Lemma D.1)}
$$

$$
= \left( \frac{5\eta_t^2}{4} + \eta_t \right) \cdot \left\| \frac{d\hat{V}_n(\pi_{\theta_t})}{d\theta_t} \right\|_2^2 \quad \left( \theta_{t+1} = \theta_t + \eta_t \cdot \frac{d\hat{V}_n(\pi_{\theta_t})}{d\theta_t} \right)
$$

$$
\leq \left( \frac{5\eta_t^2}{4} + \eta_t \right) \cdot 2 \cdot \pi_{\theta_t}(a_K)^2 \cdot (1 - \pi_{\theta_t}(a_K))^2
$$

$$
\leq \frac{9}{2} \cdot \pi_{\theta_t}(a_K)^2 \cdot (1 - \pi_{\theta_t}(a_K))^2 \quad (\eta_t \in (0, 1])
$$

$$
\leq \frac{9}{2} \cdot \pi_{\theta_t}(a_K)^2. \quad (\pi_{\theta_t}(a_K) \in [0, 1])
$$

Dividing both sides with $\pi_{\theta_{t+1}}(a_K) \cdot \pi_{\theta_t}(a_K)$, we have,

$$
\frac{1}{\pi_{\theta_t}(a_K)} - \frac{1}{\pi_{\theta_{t+1}}(a_K)} \leq \frac{9}{2} \cdot \frac{\pi_{\theta_t}(a_K)}{\pi_{\theta_{t+1}}(a_K)} \leq \frac{9}{2}. \quad (\pi_{\theta_{t+1}}(a_K) \geq \pi_{\theta_t}(a_K) > 0)
$$

Therefore, we have,

$$
\frac{1}{\pi_{\theta_1}(a_K)} - \frac{1}{\pi_{\theta_t}(a_K)} = \sum_{s=1}^{t-1} \left[ \frac{1}{\pi_{\theta_s}(a_K)} - \frac{1}{\pi_{\theta_{s+1}}(a_K)} \right]
$$

$$
= \sum_{s=1, s \in t_{\text{good}}}^{t-1} \left[ \frac{1}{\pi_{\theta_s}(a_K)} - \frac{1}{\pi_{\theta_{s+1}}(a_K)} \right] + \sum_{s=1, s \in t_{\text{bad}}}^{t-1} \left[ \frac{1}{\pi_{\theta_s}(a_K)} - \frac{1}{\pi_{\theta_{s+1}}(a_K)} \right]
$$

$$
\leq \sum_{s=1, s \in t_{\text{good}}}^{t-1} \left[ \frac{1}{\pi_{\theta_s}(a_K)} - \frac{1}{\pi_{\theta_{s+1}}(a_K)} \right]
$$

$$
\leq \sum_{s=1, s \in t_{\text{good}}}^{t-1} \left[ \frac{9}{2} \right]
$$

$$
\leq \frac{9}{2} \cdot t.
$$

In the majority of the scenarios, the parameters are initialized randomly. It means that at initialization, we have $\pi_{\theta_1}(a_K) = 1/K_{\text{EFF}}$. Once $K_{\text{EFF}}$ large enough, we have $\pi_{\theta_1}(a_K) \leq \frac{1}{c}$, for some constant $c = 11$. If $t \leq \frac{2}{9c} \cdot K_{\text{EFF}}$, then we have,

$$
\frac{1}{\pi_{\theta_t}(a_K)} \geq \frac{1}{\pi_{\theta_1}(a_K)} - \frac{9}{2} \cdot t
$$

$$
\geq \frac{1}{\pi_{\theta_1}(a_K)} \cdot \left( 1 - \frac{1}{c} \right) \geq c \cdot \left( 1 - \frac{1}{c} \right) = c - 1 \geq 10,
$$

which implies $\pi_{\theta_t}(a_K) \leq \frac{1}{10}$. Therefore, for all $t \leq \frac{2}{9c} \cdot K_{\text{EFF}}$, we have,

$$
\hat{V}_n(\pi_n) - \hat{V}_n(\pi_{\theta_t}) = (1 - \pi_{\theta_t}(a_K)) \geq 0.9 \,.
$$

The same exact proof can be done using Lemma D.3 for cluster level estimators, where $\mathcal{A}_{\text{EFF}} = \mathcal{C}$ the cluster space. $\qquad \square$

**Proposition D.5.** *Even for a single context $x$, deterministic rewards, there is problem where OPE-based learning with a linear softmax policy $\pi_\theta(a) \propto \exp(\langle\theta, h(x,a)\rangle)\mathbb{I}[a \in \mathcal{A}_{\text{EFF}}]$ can have a number of local maxima **exponential in the number of effective actions** $K_{\text{EFF}}$.*

*Proof.* Let EST an off-policy estimators considered in the paper with an action-level policy. By Lemma D.2, we have:

$$\hat{V}_n^{\text{EST}}(\pi) = \frac{1}{n}\sum_{i=1}^n \mathbb{E}_{a\sim\pi(\cdot|x_i)}\left[\hat{r}_{\text{EST},i}(a, x_i)\right], \tag{33}$$

In a single context setting, it becomes:

$$\hat{V}_n^{\text{EST}}(\pi_\theta) = \mathbb{E}_{a\sim\pi_\theta(\cdot)}\left[\frac{1}{n}\sum_{i=1}^n \hat{r}_{\text{EST},i}(a)\right], \tag{34}$$

$$= \langle\frac{1}{n}\sum_{i=1}^n \hat{r}_{\text{EST},i}, \pi_\theta\rangle. \tag{35}$$

This also holds for estimators with policies in the cluster level, as we still have:

$$\hat{V}_n^{\text{EST-C}}(\pi_\theta) = \mathbb{E}_{c\sim\pi_\theta(\cdot)}\left[\frac{1}{n}\sum_{i=1}^n \hat{r}_{\text{EST-C},i}(c)\right], \tag{36}$$

$$= \langle\frac{1}{n}\sum_{i=1}^n \hat{r}_{\text{EST-C},i}, \pi_\theta\rangle. \tag{37}$$

These softmax policies are all defined on the effective action space $\mathcal{A}_{\text{EFF}}$, be it a subset of the action space $\mathcal{A}$ or the discrete cluster space $\mathcal{C}$. Using the linearity of the objective, we can directly apply Theorem 1 from Chen et al. (2019) and obtain our result. $\qquad\square$

Finally, we also restate Proposition 3.1, and provide its proof.

**Proposition D.6.** *For an $\ell_2$ regularized (adding $\lambda||\theta||^2$, with $\lambda > 0$), linear softmax policy $\pi_\theta$, the PWLL objective $\hat{U}_n^g(\pi_\theta)$ defined as:*

$$\hat{U}_n^g(\pi) = \frac{1}{n}\sum_{i=1}^n g(R_i, \pi_0(A_i \mid X_i))\log\pi(A_i \mid X_i),$$

*is $\lambda$-strongly concave. Without regularization, the objective is concave.*

*Proof.* For any $x$ and $a \in \mathcal{A}_{\text{EFF}}(x)$, we have:

$$\pi_\theta(a|x) = \frac{\exp(\langle\theta, h(x,a)\rangle)}{\sum_{a'\in\mathcal{A}_{\text{EFF}}(x)}\exp(\langle\theta, h(x,a')\rangle)},$$

optimizing an $\ell_2$ regularized linear softmax, giving:

$$\hat{L}_n^{g,\lambda}(\pi) = \hat{U}_n^g(\pi) - \lambda||\theta||^2,$$

with $\lambda > 0$ and recall that $g \geq 0$. For strong concavity, we need to show that the Hessian $\nabla_\theta^2\hat{U}_n^g(\pi_\theta)$ is negative definite with eigenvalues bounded away from zero.

The gradient with respect to $\theta$ is: $\nabla_\theta\hat{U}_n^g(\pi_\theta) = \frac{1}{n}\sum_{i=1}^n g(R_i, \pi_0(A_i \mid X_i))\nabla_\theta\log\pi_\theta(A_i|X_i) - \lambda\theta$

For the softmax policy:

$$\nabla_\theta\log\pi_\theta(a|x) = h(x,a) - \sum_{a'}\pi_\theta(a'|x)h(x,a') = h(x,a) - \mathbb{E}_{\pi_\theta(\cdot|x)}[h(x,\cdot)]$$

Therefore: $\nabla_\theta\hat{U}_n^g(\pi_\theta) = \frac{1}{n}\sum_{i=1}^n g(R_i, \pi_0(A_i \mid X_i))\left(h(X_i, A_i) - \mathbb{E}_{\pi_\theta(\cdot|X_i)}[h(X_i,\cdot)]\right) - \lambda\theta$

Taking the second derivative: $\nabla_\theta^2 \hat{U}_n^{\mathsf{g}}(\pi_\theta) = -\frac{1}{n} \sum_{i=1}^n g(R_i, \pi_0(A_i \mid X_i)) \nabla_\theta \mathbb{E}_{\pi_\theta(\cdot|X_i)}[h(X_i, \cdot)] - \lambda I_d$, where $I_d$ is the $d \times d$ identity matrix. The gradient of the expectation is:

$$\nabla_\theta \mathbb{E}_{\pi_\theta(\cdot|x)}[h(x, \cdot)] = \sum_a \nabla_\theta \pi_\theta(a|x) h(x, a)$$

Using $\nabla_\theta \pi_\theta(a|x) = \pi_\theta(a|x)(h(x, a) - \mathbb{E}_{\pi_\theta(\cdot|x)}[h(x, \cdot)])$:

$$\nabla_\theta \mathbb{E}_{\pi_\theta(\cdot|x)}[h(x, \cdot)] = \sum_a \pi_\theta(a|x)(h(x, a) - \mathbb{E}_{\pi_\theta(\cdot|x)}[h(x, \cdot)]) h(x, a)^\top$$

This simplifies to:

$$\nabla_\theta \mathbb{E}_{\pi_\theta(\cdot|x)}[h(x, \cdot)] = \mathrm{Cov}_{\pi_\theta(\cdot|x)}[h(x, \cdot)]$$

where $\mathrm{Cov}_{\pi_\theta(\cdot|x)}[h(x, \cdot)] = \mathbb{E}_{\pi_\theta(\cdot|x)}[h(x, \cdot)h(x, \cdot)^\top] - \mathbb{E}_{\pi_\theta(\cdot|x)}[h(x, \cdot)]\mathbb{E}_{\pi_\theta(\cdot|x)}[h(x, \cdot)]^\top$

Therefore:

$$\nabla_\theta^2 \hat{U}_n^{\mathsf{g}}(\pi_\theta) = -\frac{1}{n} \sum_{i=1}^n g(R_i, \pi_0(A_i \mid X_i)) \mathrm{Cov}_{\pi_\theta(\cdot|X_i)}[h(X_i, \cdot)] - \lambda I_d$$

We can write this as: $\nabla_\theta^2 \hat{U}_n^{\mathsf{g}}(\pi_\theta) = -H - \lambda I_d$

where $H = \frac{1}{n} \sum_{i=1}^n g(R_i, \pi_0(A_i \mid X_i)) \mathrm{Cov}_{\pi_\theta(\cdot|X_i)}[h(X_i, \cdot)]$ is positive semi-definite. To see this explicitly, for any vector $v \in \mathbb{R}^d$:

$$v^\top \mathrm{Cov}_{\pi_\theta(\cdot|X_i)}[h(X_i, \cdot)]v = \mathrm{Var}_{\pi_\theta(\cdot|X_i)}[v^\top h(X_i, \cdot)] \geq 0\,,$$

with the positivity of $g$, this ensures $H$ is positive semi-definite. Then we have:

$$v^\top \nabla_\theta^2 \hat{U}_n^{\mathsf{g}}(\pi_\theta) v = -v^\top H v - \lambda v^\top v = -v^\top H v - \lambda \|v\|^2\,,$$

meaning that when $v \neq 0$, we get $v^\top \nabla_\theta^2 \hat{U}_n^{\mathsf{g}}(\pi_\theta) v \leq -\lambda \|v\|^2 < 0\,.$

This shows the Hessian is negative definite with all eigenvalues bounded above by $-\lambda < 0$. Therefore, $\ell_2$ regularized $\hat{U}_n^{\mathsf{g}}(\pi_\theta)$ is $\lambda$-strongly concave. In addition, when $\lambda = 0$, the hessian is negative semi-definite, giving simple concavity. $\qquad\square$

# E   STOCHASTIC OPTIMIZATION CONVERGENCE GUARANTEES FOR PWLL

We analyze the convergence rates of stochastic gradient methods on the PWLL objective. We formulate this as the minimization of the finite-sum objective $f(\theta) = -\hat{U}_n^g(\pi_\theta)$:

$$f(\theta) = \frac{1}{n} \sum_{i=1}^n f_i(\theta), \quad \text{where } f_i(\theta) = -g_i \log \pi_\theta(A_i \mid X_i), \tag{38}$$

where $g_i = g(R_i, \pi_0(A_i|X_i))$. We adopt the linear softmax policy parametrization in Eq. (18) with $s_\theta(x, a) = h(x, a)^\top \theta$ (lightweight parametrization in Eq. (19)). We note that our analysis extends naturally to the heavyweight parametrization in Eq. (19).

## E.1   ASSUMPTIONS AND REGULARITY

To establish problem-dependent convergence bounds, we rely on the following structural assumptions regarding the feature space and the importance weights.

**Assumption E.1** (Bounded features). For all context-action pairs $(x, a) \in \mathcal{X} \times \mathcal{A}$, the feature representations are bounded in Euclidean norm:

$$\|h(x, a)\|_2 \leq H.$$

**Assumption E.2** (Bounded weighting function). The weights $g_i = g(R_i, \pi_0(A_i|X_i))$ computed on the static dataset are strictly positive and bounded. That is, for all $i \in \{1, \ldots, n\}$:

$$0 < g_i \leq G_{\max}.$$

Assumptions E.1 and E.2 are sufficient to establish the smoothness and bounded variance of the objective $f(\theta)$. We formally derive these properties in the following proposition.

**Proposition E.3** (Regularity and Variance Bounds). *Under Assumptions E.1 and E.2, the objective $f(\theta)$ satisfies the following properties:*

1. **Global Smoothness:** *The objective is $\bar{L}$-smooth with $\bar{L} = G_{\max}H^2$.*

2. **Bounded Single-Sample Variance:** *The variance of the stochastic gradient for a single sample is bounded by $\bar{\sigma}^2 = 4G_{\max}^2 H^2$.*

3. **Bounded Mini-Batch Variance:** *For a mini-batch of size $b$, the variance is bounded by $\bar{\sigma}_b^2 = \frac{4G_{\max}^2 H^2}{b}$.*

*Proof. 1. Smoothness:* The Hessian of the objective is the weighted sum of the feature covariance matrices under the policy $\pi_\theta$:

$$\nabla^2 f(\theta) = \frac{1}{n} \sum_{i=1}^{n} g_i \text{Cov}_{a \sim \pi_\theta(\cdot|X_i)}[h(X_i, a)].$$

The spectral norm of a covariance matrix is bounded by the maximum squared norm of its random vectors. Thus, using Assumption E.1 we get that $\|\nabla^2 f(\theta)\|_{\text{op}} \leq \frac{1}{n} \sum_{i=1}^{n} g_i H^2 \leq G_{\max}H^2$.

*2. Single-Sample Variance:* We first bound the norm of the gradient for an arbitrary sample $i$. The gradient is $\nabla f_i(\theta) = -g_i(h(X_i, A_i) - \mathbb{E}_{\pi_\theta}[h(X_i, \cdot)])$. Using the triangle inequality and Assumption E.1:

$$\|\nabla f_i(\theta)\|_2 \leq g_i \left( \|h(X_i, A_i)\|_2 + \|\mathbb{E}_{\pi_\theta}[h(X_i, \cdot)]\|_2 \right) \leq G_{\max}(H + H) = 2G_{\max}H.$$

Let $\xi = \nabla f_I(\theta)$ be the stochastic gradient sampled uniformly from the dataset. The variance is bounded by the second moment:

$$\text{Var}(\xi) \leq \mathbb{E}[\|\xi\|^2] = \frac{1}{n} \sum_{i=1}^{n} \|\nabla f_i(\theta)\|^2 \leq (2G_{\max}H)^2 = 4G_{\max}^2 H^2.$$

*3. Mini-Batch Variance:* Let the mini-batch gradient be $\bar{g}_t = \frac{1}{b} \sum_{j=1}^{b} \nabla f_{i_j}(\theta)$, where indices are sampled independently with replacement. Using the standard variance reduction property for independent variables:

$$\mathbb{E}[\|\bar{g}_t - \nabla f(\theta)\|^2] = \frac{1}{b} \mathbb{E}[\|\nabla f_I(\theta) - \nabla f(\theta)\|^2] \leq \frac{4G_{\max}^2 H^2}{b}.$$

$\square$

Based on Proposition E.3, we define the following global problem-dependent constants on which our convergence rates depend:

- $\bar{L} = G_{\max}H^2$: Smoothness constant.

- $\bar{\sigma}^2 = 4G_{\max}^2 H^2$: Upper bound on the gradient variance for a single sample.

- $\bar{\sigma}_b^2 = \frac{4G_{\max}^2 H^2}{b}$: Upper bound on the gradient variance for a mini-batch of size $b$.

### E.2 CASE A: CONVEX AND SMOOTH (PWLL WITHOUT $\ell_2$ REGULARIZATION)

We begin by analyzing the standard unregularized PWLL objective. Here, the objective $f(\theta)$ is convex but not necessarily strongly convex. This implies the loss landscape may contain multiple minimizers rather than a unique global minimum. Consequently, we characterize convergence in terms of $\hat{U}_n^g(\pi_{\theta_n^{\text{opt}}}) - \hat{U}_n^g(\pi_{\bar{\theta}_T})$ (instead of $\|\theta_t - \theta_n^{\text{opt}}\|$). Here, $\theta_n^{\text{opt}} \in \arg\max_\theta \hat{U}_n^g(\pi_\theta)$ is an optimal parameter and $\bar{\theta}_T$ is the average of the SGA iterates.

**Proposition E.4.** *Let $\theta_n^{opt} \in \arg\max_\theta \hat{U}_n^g(\pi_\theta)$ be an optimal parameter. If the learning rate satisfies $0 < \eta \le \frac{1}{4L}$, then by (Garrigos & Gower, 2023, Theorem 6.9), the iterates of mini-batch SGA satisfy:*

$$\mathbb{E}\left[\hat{U}_n^g(\pi_{\theta_n^{opt}}) - \hat{U}_n^g(\pi_{\bar{\theta}_T})\right] \le \frac{\|\theta_0 - \theta_n^{opt}\|^2}{\eta T} + \frac{8\eta G_{\max}^2 H^2}{b}$$

*where $\bar{\theta}_T$ is the average of the iterates.*

Proposition E.4 highlights the trade-off inherent to constant step-size SGA: a larger $\eta$ accelerates the decay of the initial error (first term) but increases the asymptotic noise floor (second term). For a fixed horizon $T$, one can recover a convergence rate of $\mathcal{O}(1/\sqrt{T})$ by setting $\eta \propto 1/\sqrt{T}$, which balances both terms.

### E.3 CASE B: STRONGLY CONVEX AND SMOOTH (PWLL WITH $\ell_2$ REGULARIZATION)

We now move to the $\ell_2$-regularized case where the PWLL objective is strongly concave (Proposition 3.1). Precisely, we consider the regularized objective $\tilde{U}_n^\lambda(\theta) = \hat{U}_n^g(\pi_\theta) - \frac{\lambda}{2}\|\theta\|^2$.

Strong convexity implies the existence of a unique global minimizer. This allows us to guarantee convergence of the parameters $\theta_t$ themselves, which is a stronger condition than value convergence.

**Proposition E.5.** *Let $\theta_{n,\lambda}^{opt} = \arg\max_\theta \tilde{U}_n^\lambda(\theta)$ be the unique optimal parameter. If the learning rate satisfies $0 < \eta \le \frac{1}{2(G_{\max}H^2 + \lambda)}$, then by (Garrigos & Gower, 2023, Theorem 6.12):*

$$\mathbb{E}\left[\|\theta_t - \theta_{n,\lambda}^{opt}\|^2\right] \le (1 - \eta\lambda)^t \|\theta_0 - \theta_{n,\lambda}^{opt}\|^2 + \frac{8\eta G_{\max}^2 H^2}{\lambda b}$$

The regularized case demonstrates a convergence rate that is significantly faster than the rate of the unregularized case.

## F ADDITIONAL EXPERIMENTS

### F.1 DETAILED EXPERIMENTAL SETTING

Table 1: Statistics of Post Processed Datasets

| Dataset | Num. of actions | Num. of samples |
|---------|-----------------|-----------------|
| MovieLens | $60,000$ | $132,744$ |
| Twitch | $200,000$ | $400,000$ |
| GoodReads | $1,000,000$ | $400,000$ |

**Experimental Setting.** Our experimental setup is designed to study the behavior of the different policy learning paradigms in large action spaces. To this end, we use three large action spaces collaborative filtering datasets: Movielens (Lam & Herlocker, 2016), Twitch (Rappaz et al., 2021) and GoodReads (Wan et al., 2019) that are preprocessed to obtain a user-item interaction matrix[3]. We follow the exact procedure of Sakhi et al. (2023b) to pre-process the datasets. The statistics

---

[3]Code and datasets are heavy, both will be released upon acceptance.

of the obtained datasets are described in Table 1. For each user, we keep half of its history as the context $x$, and use the other half of the history as the products with positive reward, which align the learned policies to recommend new and relevant items. We direct the interested readers to Sakhi et al. (2023b) for a detailed description of the experimental setup.

The large action space scenario restricts the policies used to the inner product parametrization (Aouali et al., 2022). This parametrization is essential to leverage Maximum Inner Product Search algorithms (Shrivastava & Li, 2014) for fast query response. In particular, we adopt policies of the following form:

$$\pi_\theta(a|x) \propto \exp(\langle h_\Gamma(x), \beta_a \rangle) \, ,$$

with the learnable parameter $\theta = [\Gamma, \beta]$, $h_\Gamma : \mathcal{X} \to \mathbb{R}^\ell$ defines the context embedding function in $\mathbb{R}^\ell$ and $\beta$ the actions embeddings of size $K \times \ell$. In all our experiments and unless it is explicitly stated, we follow the procedure of Sakhi et al. (2023c) to define our policies. We start by extracting action embeddings $\beta_0$ using an SVD decomposition of the user-item matrix. These embeddings help us define the context embedding function $h_\Gamma$ and our logging policy $\pi_0$. $h_0$ is set to the average embeddings of the observed actions in the contexts and is fixed for the logging policy $\pi_0$. Using the SVD action embeddings $\beta_0$, we define our logging policy $\pi_0$ as:

$$\pi_0(a|x) \propto \exp\left(\frac{1}{t} \langle h_0(x), \beta_{0,a} \rangle\right) \mathbb{I}\left[a \in \text{TOP}^{k_0}(x)\right] \, ,$$

with $t$ the temperature of the logging policy, and $k_0$ define the support of the logging policy, concentrating on the top $k_0$ actions with: $\text{TOP}^{k_0}(x) = \text{argsort}_{a_1, \cdots, a_{k_0}} \langle h_0(x), \beta_{0,a} \rangle$.

If not explicitly stated, $k_0$ is set to 100 and the temperature at $t = 1$ in all experiments. This policy is used to collect the offline dataset $\mathcal{D}_n = \{X_i, A_i, R_i\}_{i \in [n]}$ on which all trainings are conducted. For each $i \in [n]$ in the processed dataset, $X_i$ is the user history, $A_i$ is the action played by the logging policy $\pi_0(\cdot|X_i)$ and $R_i = \mathbb{1}[A_i \in H_i]$ the observed reward, which is if the action played is in the hidden items of user $i$.

**Trained Policies Parameterizations.** We adopt two parameterizations of the trained policies. The first one is a **heavyweight** parametrization, and focuses on learning the embeddings of the actions $\beta$ (be it $\mathcal{A}$ of size $K$ or $\mathcal{C}$ of size $|\mathcal{C}|$), meaning that $\theta$ in this case is $\beta$. For action-level policies, this gives $\beta \in \mathbb{R}^{K \times \ell}$ and for any $x$:

$$\pi_\beta(a|x) = \frac{\exp(\langle h_0(x), \beta_a \rangle)}{\sum_{a' \in \mathcal{A}_{\text{EFF}}(x)} \exp(\langle h_0(x), \beta_{a'} \rangle)} \, ,$$

with $\mathcal{A}_{\text{EFF}}(x) \subset \mathcal{A}$, which depends on the choice of the practitioner, for example $\mathcal{A}_{\text{EFF}}(x) = S_0(x)$, the support of $\pi_0$ for context $x$ when we optimize IPS objectives. For cluster-level policies, this gives a $\beta \in \mathbb{R}^{|\mathcal{C}| \times \ell}$ and for any $x$:

$$\pi_\beta(c|x) = \frac{\exp(\langle h_0(x), \beta_c \rangle)}{\sum_{c' \in \mathcal{C}} \exp(\langle h_0(x), \beta_{c'} \rangle)} \, .$$

This is used by default if nothing is explicitly stated.

We have also define a **lightweight** parametrization, where only a small projection $W \in \mathbb{R}^{\ell \times \ell}$ is learned, giving in action level policies:

$$\pi_W(a|x) = \frac{\exp(\langle h_0(x)W, \beta_{a,0} \rangle)}{\sum_{a' \in \mathcal{A}_{\text{EFF}}(x)} \exp(\langle h_0(x)W, \beta_{a',0} \rangle)} \, ,$$

using $\beta_0$, the embeddings of $\pi_0$. For cluster level policies, we first define $\bar{\beta}_0 \in \mathbb{R}^{|\mathcal{C}| \times \ell}$ with $\bar{\beta}_{0,c} = \frac{1}{|c|} \sum_{a \in c} \beta_{0,a}$, and use it to define the cluster level policy:

$$\pi_W(c|x) = \frac{\exp(\langle h_0(x)W, \bar{\beta}_{c,0} \rangle)}{\sum_{c' \in \mathcal{C}} \exp(\langle h_0(x)W, \bar{\beta}_{c',0} \rangle)} \, .$$

**Reward Model.** The reward model used $\hat{r}$ is learned using regularized linear regression the collected interaction data, with $\hat{r}(x, a) = \langle h(x), \theta_a \rangle$.

**Clustering and $\epsilon$ used.** We use the embeddings $\beta_0$, combined with K-means clustering to find our clusters. The number of clusters is set to 2000 for all datasets and experiments. For `PC`, the $\ell_2$ threshold $\epsilon$ is set to 0.1.

### F.2 Additional results

**Training progress using two different parametrizations.** Fig. 5 shows the training progress over 10 epochs on all three datasets, comparing heavyweight vs. lightweight policies.

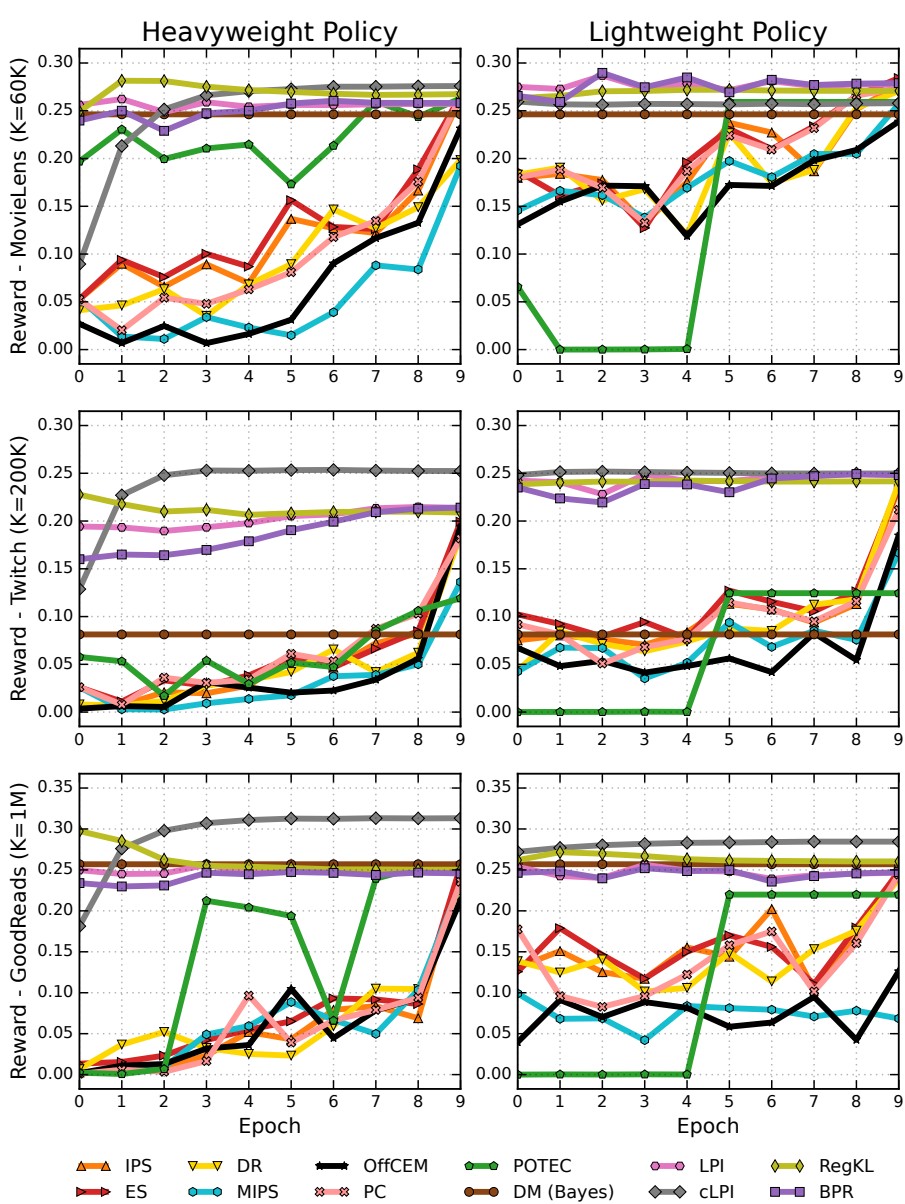

Figure 5: Training progress over 10 epochs on all three datasets, comparing heavyweight vs. lightweight policies.

**Benefits of objective-aware parametrization.** Fig. 6 shows the effect of objective-aware policy parameterizations for two different objectives and three large action space datasets.

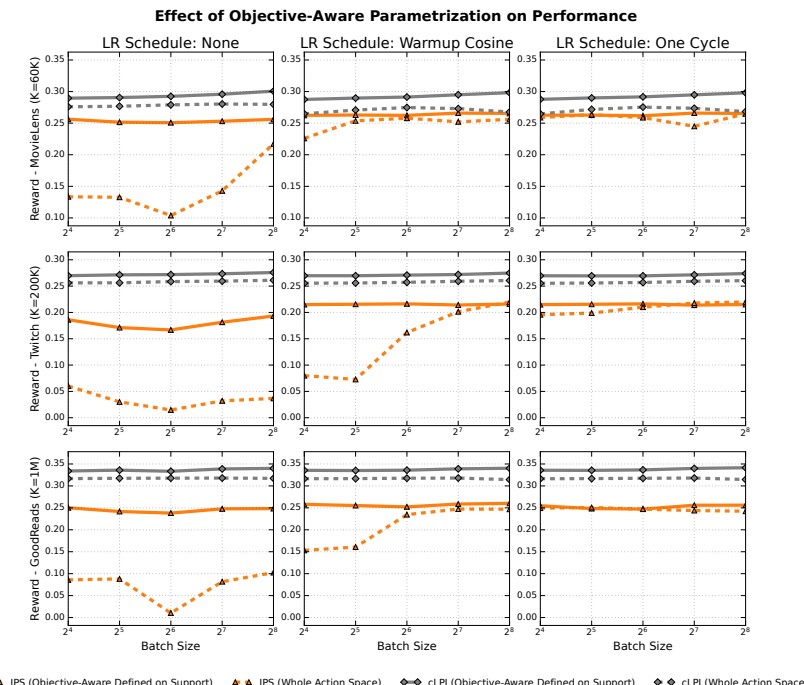

Figure 6: The effect of objective-aware parametrization for `IPS` and `cLPI` on three large-scale datasets

**Average MSE.** Fig. 7 shows the average MSE by dataset and method. Several methods are excluded from the figure, as their high MSE values would distort the scale and obscure the comparison.

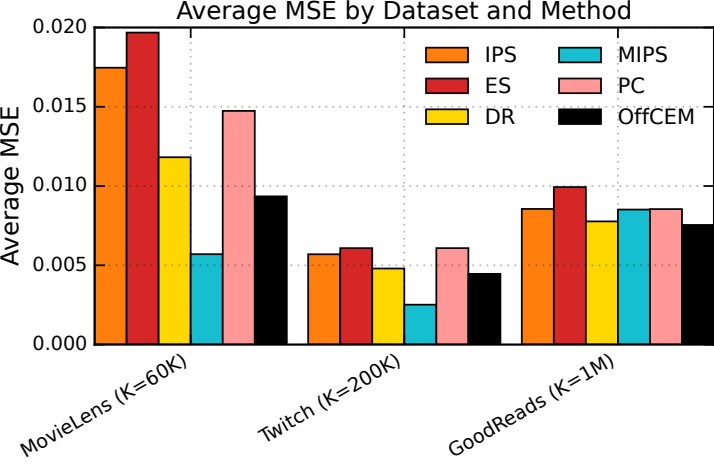

Figure 7: Average MSE by Dataset and Method. Several methods are excluded from the figure, as their high MSE values would distort the scale and obscure the comparison.

**MSE progress during training.** Figs. 8a to 8c show the progress of the MSE over 10 epochs on all three datasets. Several methods are excluded from the figure, as their high MSE values would distort the scale and obscure the comparison.

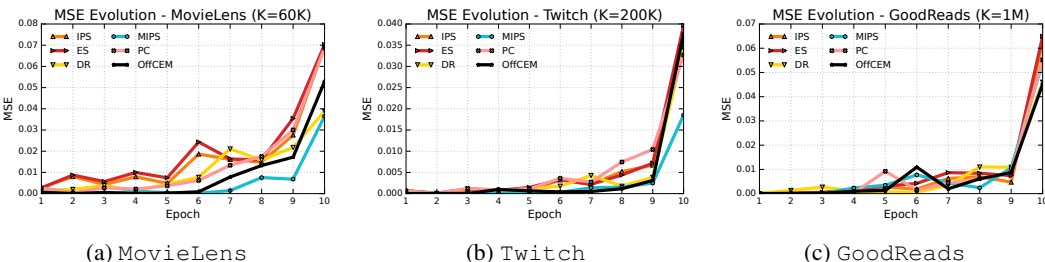

(a) MovieLens    (b) Twitch    (c) GoodReads

Figure 8: MSE progression over 10 epochs across datasets.

## F.3 RESULTS AVERAGING DIFFERENT SEEDS

In this experiment, we analyze the reward evolution of representative PWLL and OPE-based methods on the three considered datasets. We compare two distinct optimization configurations: (i) a standard off-the-shelf Adam optimizer, and (ii) a carefully tuned setup using Adam with an optimized batch size and a one-cycle learning-rate scheduler. This comparison enables us to isolate the effect of optimization on stability and convergence. Each method is evaluated over 5 random seeds, and we report the mean reward along with a shaded standard deviation region to visualize sensitivity to optimization randomness.

In Figure 9, across all datasets and optimization settings, we observe that OPE-based methods (cIPS, IX, and even POTEC) not only reach inferior performance but also suffer from considerably higher variance. Their uncertainty bands are significantly wider, indicating unstable optimization. In contrast, PWLL-based methods exhibit near-invisible variance bands, with standard deviations roughly an order of magnitude smaller on average.

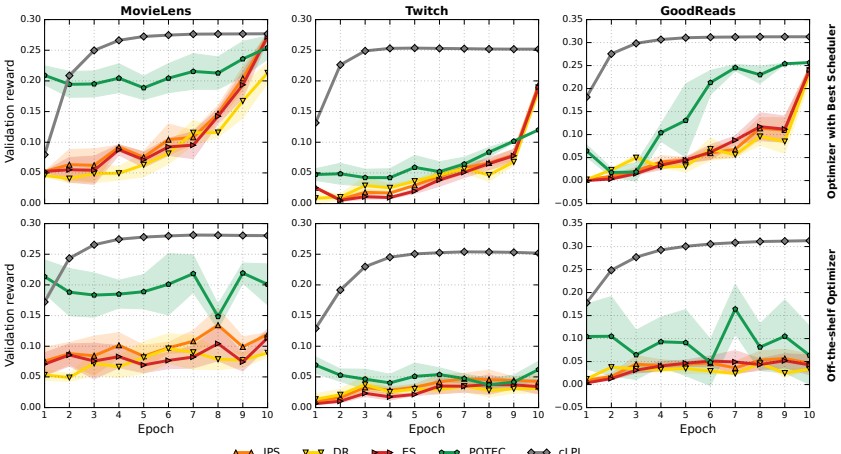

Figure 9: cLPI vs OPE-Based methods: Evolution of rewards averaged over 5 different seeds. cLPI is more stable to optimize and reaches better policies.

Finally, in Figure 10, we observe that adopting an Objective-Aware parametrization yields further performance and stability improvements. For example, cIPS with Objective-Aware parametrization surpasses cIPS while maintaining lower variability, and cLPI in its Objective-Aware form consistently achieves the best overall performance. These results demonstrate that the combination of PWLL objectives and clever parametrization leads to more robust and more effective learned policies, while being very simple to implement.

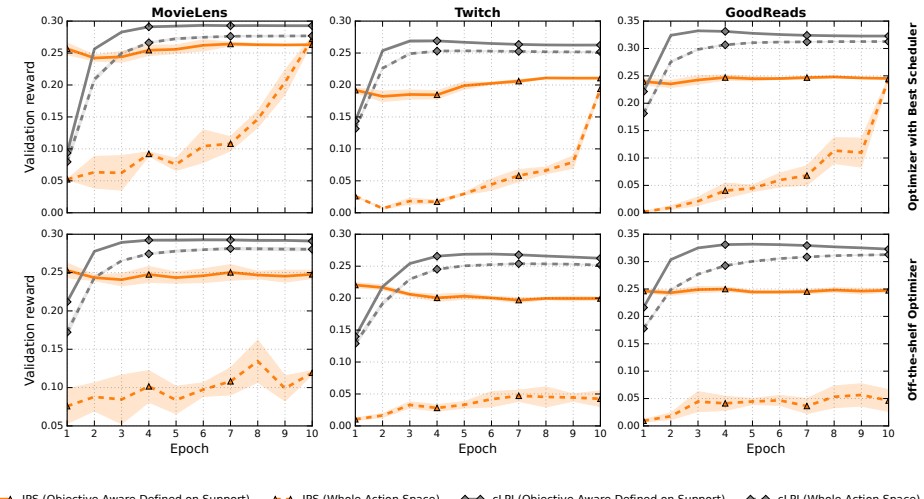

Figure 10: Objective Aware Parametrisation: Evolution of rewards averaged over 5 different seeds. Objective Aware Parametrization stabilizes and improves performance for PWLL and OPE methods.

### F.4  ABLATION - SENSITIVITY TO REWARD NOISE

In the original evaluation setup (see Appendix F), the observed reward is deterministic; for user $i$, we have $R_i = \mathbb{1}[A_i \in H_i]$, meaning that a positive reward is returned only when the selected action belongs to the user's hidden set $H_i$. In this section, we investigate robustness to reward noise by introducing stochasticity in the form:

$$R_i \sim \mathbb{1}[A_i \in H_i]\,(1 - B(\epsilon)) + B(\epsilon)\,s,$$

where $B(\epsilon)$ is a Bernoulli random variable with parameter $\epsilon$, and $s \in [0, 1]$ is a shift. This results in noisy rewards supported on $[0, 1]$. Note that any reward scaling can be normalized to this range via $R/R_{\max}$ when $R_{\max} > 1$.

We evaluate six configurations defined by noise parameters $\epsilon \in \{0.1, 0.2, 0.3\}$ and reward shifts $s \in \{0, 0.5\}$. All methods are trained using the best-performing optimization schedule (one-cycle) to isolate the effect of noise. Results are reported in Fig. 11.

We observe that increasing the noise level when $s = 0$ consistently harms all methods, as expected from a more stochastic reward signal. In contrast, when $s = 0.5$, higher noise tends to increase the overall reward level, since the shift raises the baseline reward. Across all noise–shift conditions, PWLL-based objectives maintain a clear advantage over OPE-based methods. When $s = 0$, `RegKL` and `cLPI` perform similarly, confirming that both benefit from the logarithmic reparameterization. However, as both noise and shift increase, `RegKL` begins to outperform `cLPI`, suggesting that, with an appropriately chosen regularization weight $\beta$, `RegKL` remains highly competitive even under reward high stochasticity.

**Conclusion.** PWLL methods demonstrate robustness to reward noise, leading to improved stability and performance compared to traditional OPE-based objectives, even in challenging noise regimes.

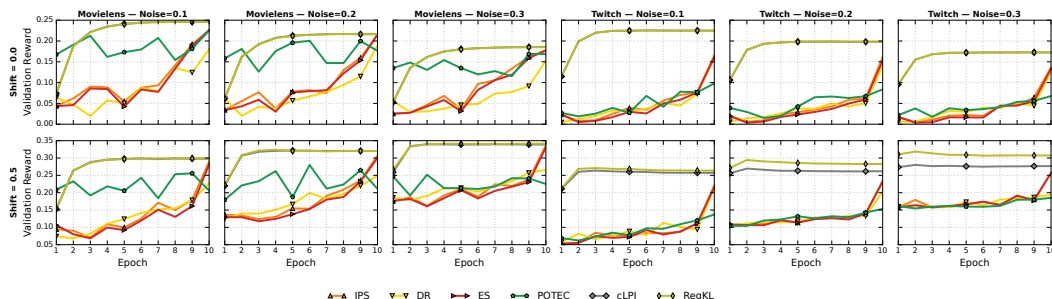

Figure 11: Ablation - Sensitivity to reward noise

## F.5 ABLATION STUDY ON HYPERPARAMETERS AND LOG TRANSFORM

In this section, we evaluate the impact of hyperparameter choices on cIPS, cLPI, RegKL, and RegKL-LIN (the non-logarithmic variant of RegKL). All methods are run using the best-performing optimization configuration (optimizer + learning rate scheduler), ensuring that differences are driven solely by hyperparameter values and by whether the policy transformation is linear or logarithmic. The results are shown in Figure 12.

- **cIPS** consistently fails to reach competitive performance across all values of $\tau$, especially in large action spaces. Its PWLL counterpart, cLPI, dominates for every $\tau$, converging faster and achieving superior results.

- For the KL-based objectives, we restrict to $\beta \geq 0.1$ in order to avoid numerical instability from the exponential term ($\exp(1/\beta) > 2 \cdot 10^5$ for $\beta < 0.1$). The same trend is observed: the PWLL variant (RegKL) reliably outperforms its linear analogue (RegKL-LIN) across all $\beta$, exhibiting more stable training dynamics, faster convergence and better performance.

**PWLL dominates.** Across both objective families, replacing linear weights with *log-transformed* policy weights (PWLL) consistently provides **greater robustness to hyperparameters**, **faster optimization**, and **higher final performance**, even more in challenging large-action-space settings.

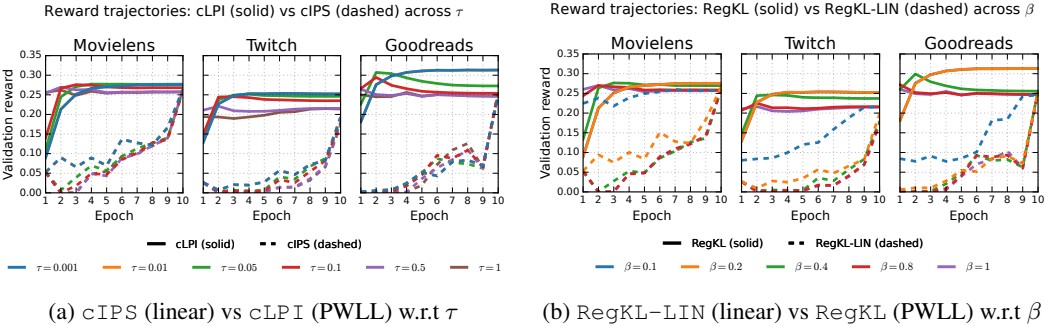

(a) cIPS (linear) vs cLPI (PWLL) w.r.t $\tau$

(b) RegKL-LIN (linear) vs RegKL (PWLL) w.r.t $\beta$

Figure 12: Ablation Study on hyper-parameters and Log Transform

## F.6 ABLATION: PWLL IN SMALLER ACTION SPACES

We have shown that PWLL provides a more benign optimization landscape and yields stronger policies than OPE-based objectives in large action spaces. Here, we examine whether these benefits also extend to smaller action spaces. We construct a reduced version of Movielens by subsampling the action space to $K \in \{100, 500, 1000, 5000\}$ items.

Figure 13 reports performance across varying $K$ for cIPS (linear) and its PWLL-enhanced counterpart cLPI (log). In small action space settings ($K \leq 500$), cIPS convergences faster than cLPI, but cLPI identifies a better maxima by the end of the 10 epochs. For medium action spaces

$(K \geq 500)$, `cLPI` consistently outperforms `cIPS`, converging faster and identifying a better maximum. These results indicate that the optimization advantages of PWLL can still be beneficial in medium sized action space settings.

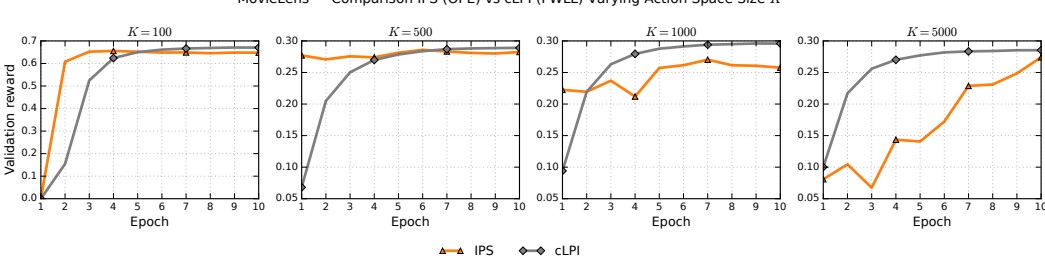

Figure 13: PWLL (`cLPI`) vs OPE (`IPS`) in Smaller Action Spaces.

### F.7 ABLATION - SENSITIVITY TO THE NUMBER OF CLUSTERS

In this study, we compare our simple PWLL objective `cLPI` against `MIPS` and `POTEC`, two more complex OPE-based methods specifically designed for large action spaces. These baselines rely on a clustering function to reduce variance, and `POTEC` additionally leverages a reward model $\hat{r}$. We examine how the number of clusters affects their optimization performance. Figure 14 reports the results.

`POTEC` generally outperforms `MIPS` for all numbers of clusters. However, both methods exhibit optimization instability across settings. While `POTEC` can occasionally match the final performance of `cLPI` on Movielens for a carefully selected number of clusters (1000), it consistently falls short on Twitch regardless of the cluster configuration.

**Conclusion.** These findings demonstrate that focusing on optimization properties pays off: despite its simplicity, the PWLL objective `cLPI` can consistently outperform intricate OPE-based approaches tailored to large action spaces, even with the best finetuning.

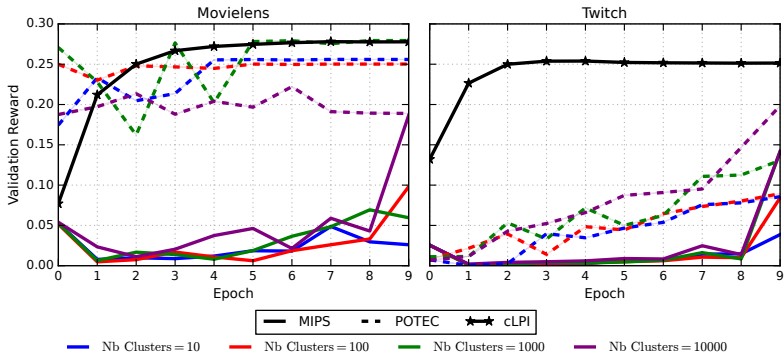

Figure 14: PWLL (`cLPI`) vs `POTEC` and `MIPS`, changing the number of clusters.

### F.8 ABLATION - DIFFERENT LOGGING SUPPORTS

We conduct experiments to quantify how increasing or restricting the support of the logging policy affects policy learning, comparing PWLL and OPE-based methods. Figure 15 compiles the results and show that PWLL is still better than OPE based approaches for different logging support sizes.

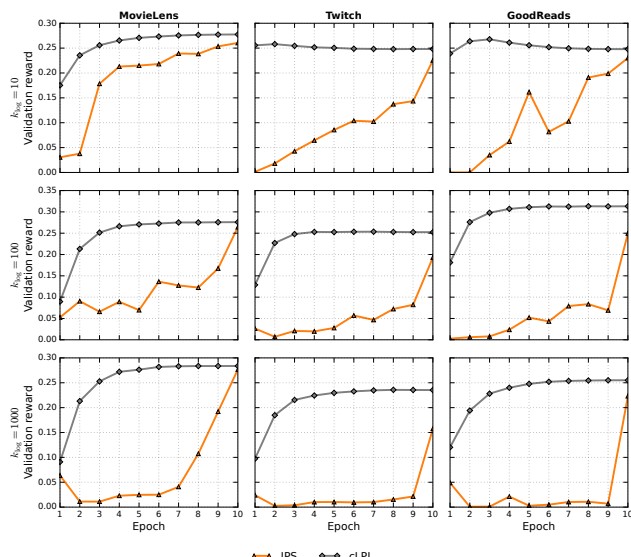

Figure 15: PWLL vs OPE: Different Logging Support sizes $k_{\log}$

## F.9 Pessimism Does not Solve Optimization Problems

Pessimism in face of uncertainty Jin et al. (2021) is motivated through a pure statistical learning rationale and is used to provide better statistical guarantees and more controlled excess risk. In the context of OPL, pessimistic strategies are derived combining concentration bounds with class complexity measures, be it VC dimension (Swaminathan & Joachims, 2015a) or PAC-Bayesian tools (London & Sandler, 2019; Aouali et al., 2023; Sakhi et al., 2024). For example, in its PAC-Bayesian formulation, the pessimistic objectives are all written in the following form:

$$\arg\max_{\pi_\theta} \quad \hat{V}_n(\pi_\theta) - \frac{\lambda}{n}||\theta - \theta_0||_2^2 \,,$$

Adding an $\ell_2$ regularization term that pulls the parameters $\theta$ towards the behavior policy parameters $\theta_0$ (defining $\pi_0$) induces pessimism by encouraging the learned policy to stay close to $\pi_0$ in parameter space. However, the optimization landscape of this objective becomes concave only when the regularization weight $\lambda$ is sufficiently large for the $\ell_2$ term to dominate. In that regime, the objective is indeed easier to optimize, but becomes overly conservative, yielding policies that remain too close to $\pi_0$ and under-exploit potential improvements. Figure 16 confirms this empirically: pessimistic approaches, whether based on Sample Variance Penalisation (SVP) (Swaminathan & Joachims, 2015a), PAC-Bayesian learning with clipped IPS (London & Sandler, 2019), Exponential Smoothing (Aouali et al., 2023), or Logarithmic Smoothing (Sakhi et al., 2024), fail to outperform cLPI for any value of $\lambda$.

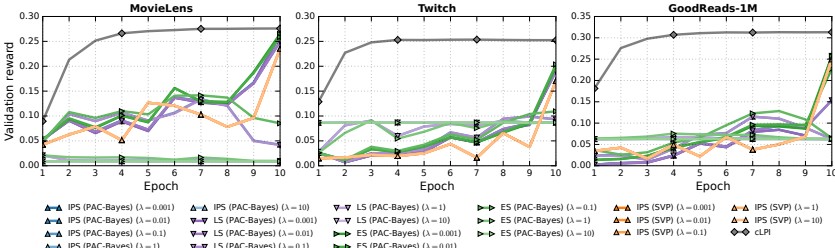

Figure 16: cLPI outperforms the pessimistic approaches. Some methods do not appear in the plot because their curves overlap.

## LLM USAGE

We acknowledge the use of Large Language Models (LLMs) for writing assistance in the preparation of this manuscript. Their role was strictly limited to improving sentence-level grammar, phrasing, and readability. The core scientific ideas, experimental design, analysis, and conclusions presented herein are entirely the work of the authors.

