# OpenReview forum: "Off-Policy Learning in Large Action Spaces: Optimization Matters More Than Estimation"
_ICLR.cc/2026/Conference — Submitted to ICLR 2026_

### Official Review · Reviewer_u2kL · 2025-10-17

**Soundness:** 4
**Presentation:** 3
**Contribution:** 3
**Rating:** 6
**Confidence:** 3

**Summary:**

The paper argues that in offline contextual bandits with very large action spaces, the main bottleneck isn’t statistical estimator quality but the optimization landscape induced by common OPE-driven objectives (IPS/DR and large-action variants). It shows these objectives become highly non-concave and plateau-prone as the effective action size grows, and that “estimator-aware” policy parameterizations (e.g., POTEC’s cluster-level policy) help but don’t fix the root issue. In response, the authors propose PWLL objectives that trade value-estimation fidelity for concave (often strongly concave) optimization under linear-softmax policies, leading to more robust learning and competitive, or superior, policies in large-K settings. Empirical results on large recommender datasets support this optimization-first view.

**Strengths:**

Reframes OPL progress from “better estimators => better policies” to “better optimization => better policies” an interesting perspective shift.


Connects asymptotic solutions of OPE estimators to policy-class design (objective-aware parametrization), giving clear recipes (e.g., restrict to $S_0(x)$, cluster-level optimization).


Advocates and analyzes a PWLL family that prioritizes concave landscapes, unifies/contrasts several known surrogates

Comprehensive experiments across three large-K recommenders, stress-tests with batch-size and LR-schedule sweeps strengthen the optimization narrative.

Clear comparison showing POTEC’s gains stem from optimization structure rather than estimator improvements.

**Weaknesses:**

The concavity guarantee for PWLL holds for linear softmax policies, but many real systems use deeper, non-linear policies or two-tower networks. It’s unclear how much of PWLL’s optimization advantage persists beyond this setting or whether local strong concavity emerges in practice.


While the paper argues estimation fidelity is overrated for learning, some readers will want generalization or regret bounds for PWLL (even if biased as an estimator). e.g., performance guarantees vs. the best policy in a class under support deficiencies or misspecified reward noise.

“Reward” is reported but the construction of rewards in the static datasets (implicit feedback, scaling, propensity correction, negative sampling) is not deeply detailed in the main text. Since PWLL weights use $R_i$ (and sometimes $\exp(R_i/\beta)$), reward scale/noise critically affects training stability, so more robustness checks (e.g., rescaling/noise injection/label smoothing) would help.

**Questions:**

Since LPI and especially RegKL use $R_i$ inside $g(\cdot)$ or $\exp(R/\beta)$, how sensitive are results to rescaling or label noise? A controlled study varying reward distributions and the temperature $\beta$ would help operationalize the recipes.

PWLL resembles advantage-weighted/behavior-regularized policy updates in offline RL (e.g., CRR/AWAC). A brief comparison (objectives, weighting, guarantees) would position your contribution within a broader literature and might suggest cross-pollination.

---

> ### Author Response · Authors · 2025-11-23
>
> We sincerely thank the reviewer for their thoughtful positive feedback. We are pleased that you found our optimization-first perspective novel, our objective-aware parametrization clear, and our experimental evaluation comprehensive. Below, we detail our responses and summarize the key revisions.
>
> ## **Summary of changes**
>
> * Added a **new ablation study** on reward scaling and label noise and shift (Figure 11 in Appendix F.4).
> * Added a discussion on **non-linear (deep) policies** (Remark 3.2).
> * Added discussion on RL methods (lines 398-406).
>
> ---
>
> ### **1) Robustness to reward scale and noise**
>
> LPI and cLPI are linear in the reward: multiplying all rewards by a positive constant only rescales the gradient. Thus, reward-scaling ablation is equivalent to a learning-rate ablation for these two methods. Figure 2 shows that they are robust to varying learning-rate variations, and hence to varying reward scale.
>
> For RegKL, scaling the reward is equivalent to scaling the coefficient $\beta$. Thus, changing the reward scale can be absorbed by redefining $\beta$. Thus, the corresponding ablation reduces to varying $\beta$: which we did in Figure 12(b), showing robustness.
>
> Reward noise, however, changes the effective learning signal. For this reason, we additionally include experiments evaluating robustness to noisy reward feedback (Figure 11, Appendix F.4). PWLL objectives are affected by reward noise, but they remain stable, and continue to outperform OPE objectives even under noisy rewards.
>
> ---
>
> ### **2) Beyond linear-softmax policies**
>
> You are correct that our global concavity guarantee applies to linear softmax policies. We now clarify how these insights extend to deeper networks.
>
> 1.  **Frozen Encoder (used in recommender systems):** In many large-scale systems (e.g., two-tower recommenders), a deep encoder is pre-trained and *frozen*, and only a final *linear head* is trained for the specific task. In this case, our guarantees hold, as the optimization is only over the parameters of this linear head.
>
> 2.  **Fully Trainable (End-to-End):** For fully trainable networks, global concavity no longer holds. However, PWLL’s gradient $\nabla_\theta \log \pi_\theta(A_i|X_i)$ is similar to that of the cross-entropy loss, which is known for stable deep optimization. Thus, despite relaxed formal guarantees, the practical advantages remains.
>
> ---
>
> ### **3) Connection to RL**
>
> We are thankful for the reviewer for bringing these RL work to our attention. Below we position PWLL within the broader RL literature:
>
> **Setting and motivation.** We focus on offline contextual bandits with large action spaces. Our goal is to identify and analyze objectives and policy parametrizations that remain optimizable at scale. In contrast, AWAC/CRR are designed for multi-step offline RL, where the primary motivation is to handle distributional shift and bootstrapping, not large-$K$ policy optimization pathologies.
>
> **Perspective.** We bring optimization to the attention of the community, and show that simple PWLL objectives and objective-aware parametrizations outperform sophisticated OPE objectives in large action spaces.
>
> **Architectural differences.** AWAC/CRR rely on a learned critic and advantages $A_\phi$. In contrast, PWLL is:
>
> * **Critic-free.** Operates without learned value functions or advantage estimates.
> * **Directly uses data.** Uses a weighting function $g(R_i,\pi_0(A_i\mid X_i))$ that depends only on observed rewards and logging propensities, bypassing value-function approximation.
>
> This design prioritizes simplicity and stability in bandit settings, and our results show that, **in the offline contextual bandit setting** substantial gains are already achievable without the complexity of critic learning.
>
> Conceptually, PWLL extends naturally to multi-step RL but a careful treatment of this extension is left for future work.
>
> ---
>
> ### **4) On generalization bounds**
>
>
> PWLL can be cast as weighted empirical risk minimization. Thus, with bounded weights $|g| \le G_{max} $ and bounded features $||h(x,a)|| \le B $, standard Rademacher arguments give an $O(1/\sqrt{n})$ uniform deviation, so PWLL converges to the maximizer of its own expectation. This does not imply consistency for the true value $V(\pi)$; the bias does not vanish. However, our paper shows that in large-$K$ regimes, this bias is negligible compared to the optimization error of OPE objectives, whose landscapes become highly non-concave and difficult to train. PWLL remains concave and easy to optimize, so its total error (bias + optimization) is substantially smaller as we showed in our experiments.
>
> ---
>
> Thank you again for the thoughtful and positive feedback. We hope the clarifications and additional experiments fully address your comments, and would greatly appreciate your consideration in raising your support for our paper.

---

### Official Review · Reviewer_Vgm9 · 2025-10-24

**Soundness:** 3
**Presentation:** 3
**Contribution:** 2
**Rating:** 4
**Confidence:** 4

**Summary:**

This paper argues that in off-policy learning (OPL) for large action spaces, the main challenge lies in optimization, not in improving the statistical accuracy of off-policy evaluation (OPE) estimators. Through theoretical analysis and large-scale experiments, the authors show that common OPE-based objectives (e.g., IPS, DR, MIPS, OffCEM) have highly non-concave landscapes that hinder learning, even when the estimators are accurate.

They propose two remedies: (1) objective-aware policy parametrization, aligning the policy’s structure with the estimator’s bias to ease optimization, and (2) policy-weighted log-likelihood (PWLL) objectives, which are concave and easy to optimize but still yield strong policies.

Experiments on large recommendation datasets demonstrate that these simpler, optimization-focused methods outperform complex OPE-based approaches. The key takeaway is that optimization-friendly objectives lead to better policies than statistically precise but hard-to-train estimators.

**Strengths:**

- The paper challenges the dominant estimator-centric paradigm in off-policy learning (OPL), emphasizing that optimization bottlenecks—not estimation errors—are the main limiting factor in large action spaces. This conceptual reframing is original and relevant to both theory and practice.

- Strong theoretical foundation: The authors provide rigorous asymptotic analysis showing how common off-policy evaluation (OPE) objectives induce difficult non-concave landscapes and multiple local optima, particularly as action space grows. The derivation of objective-aware parametrization and analysis of concavity in PWLL objectives is theoretically sound.

- Clear algorithmic proposals: The introduction of Policy-Weighted Log-Likelihood (PWLL) objectives is elegant and well-motivated. These objectives are shown to yield strongly concave optimization problems under linear softmax policies, leading to simple yet effective learning procedures.

- Comprehensive empirical study: Experiments on large-scale datasets (MovieLens, Twitch, GoodReads) convincingly demonstrate that PWLL-based methods (e.g., cLPI, RegKL) outperform complex OPE-based baselines (e.g., OffCEM, POTEC) in both stability and final reward, validating the paper’s core claim.

- Practical insights and implications: The work provides guidelines for large-scale OPL design—notably the benefits of objective-aware and lightweight parametrizations—offering direct takeaways for practitioners.

**Weaknesses:**

- Limited novelty in algorithmic form: While the conceptual shift toward optimization-centric design is insightful, the specific PWLL objectives (LPI, cLPI, RegKL) closely resemble known reward-weighted or behavior-cloning-style objectives. The paper’s novelty lies more in perspective than in methodology.

- Limited theoretical depth on optimization dynamics: Although the paper states concavity results, the theoretical section does not deeply analyze gradient dynamics or convergence rates under stochastic optimization—key for understanding large-scale practicality.

- Scope of baselines for theoretical analysis and experiment: The study omits recent pessimistic or PAC-based OPL methods (e.g., conservative or uncertainty-aware learning) that might also improve optimization robustness. Including them could strengthen the empirical claims.

- Clarity and positioning: The framing could better articulate how PWLL fits within or generalizes existing frameworks like reward-weighted regression or entropy-regularized policy gradients, to avoid appearing incremental.

**Questions:**

- What does $\hat{r}$ mean in the asymptotic regime in Eq (10)? Doesn't $\hat{r}$ become $r$?
- Trade-off analysis: How does the loss in value estimation accuracy quantitatively relate to the gains in optimization stability? Is there a measurable regime where OPE objectives outperform PWLL?

- Generalization to reinforcement learning: Can the optimization-centric insights transfer beyond contextual bandits to full RL settings where multi-step credit assignment exists?

- Robustness: How does PWLL perform under misspecified reward signals or biased logged data (e.g., when π₀ has deficient support)?

---

> ### Author Response · Authors · 2025-11-23
> **Part I**
>
> We thank the reviewer for the detailed assessment and for highlighting the strengths of our theoretical analysis, algorithms, and large-scale experiments.
>
> Below, we first summarize the changes made to the paper and then address your concerns point by point.
>
> ### **Summary of changes**
>
> * Positioning and the bandit-RL connection (lines 398-407).
> * Added optimization convergence guarantees (lines 306–311 and Appendix E).
> * Added pessimistic baselines (Figure 16, Appendix F.9).
> * Added small-$K$ experiments clarifying when OPE is preferable (Figure 13, Appendix F.6).
> * Added ablations for noisy rewards and logging policies with deficient support (Figure 11, Appendix F.4; Figure 15, Appendix F.8).
>
> ---
>
> ### **1. Novelty and positioning**
>
> We shed light on optimization as the fundamental challenge in **large-$K$ offline contextual bandits**, and from this analysis a **general PWLL family** and **objective-aware policy parametrizations** naturally emerge. The novelty lies in four concrete contributions:
>
> * **1. Exposing optimization as the primary bottleneck at large $K$.** We show theoretically (Props. 2.1-2.2) and empirically (Fig. 2) that OPE objectives induce severely ill-conditioned landscapes, making optimization, not estimation error, the dominant failure mode.
>
> * **2. Deriving objective-aware parametrizations.** Using asymptotic solutions, we obtain principled parametrizations that drastically improve trainability (Fig. 3):
>
>   * restrict $\mathcal A_{\text{eff}}(x)=S_0(x)$ for IPS/cIPS/ES;
>   * use POTEC-style two-stage parametrization for OffCEM/MIPS/PC.
>
> * **3. Unifying PWLL as an optimization-friendly family.**
> With optimization as the design goal, the PWLL family emerges naturally: it enjoys **provable concavity** (Prop. 3.1), is straightforward to optimize at scale, and **recovers the same asymptotic solutions** as several OPE objectives (e.g., cLPI mirrors cIPS). This unifies a range of heuristics under a single optimization-oriented framework.
>
> * **4. Large-scale empirical validation.**
> We provide experiments at $K=10^6$ with extensive ablations, demonstrating clearly that simple PWLL objectives consistently outperform sophisticated OPE objectives.
>
> ---
>
> ### **2. Optimization dynamics and convergence rates**
>
> Since PWLL is **concave** (unregularized) and **strongly concave** with $\ell_2$ regularization (Prop. 3.1), convergence reduces to verifying **$L$-smoothness** and **bounded gradient variance**. Under $||h(x,a)||\le H$ and $g(R_i, \pi_0(A_i|X_i)) \le G_{\max}$, PWLL is $\bar{L}$-smooth where $\bar{L} = G_{\max} H^2,$ and its gradient variance are bounded by $ G_{\max}^2 H^2$. This allows us to apply standard SGA results from [1].
>
> * **PWLL without $\ell_2$ regularization (concave and $\bar{L}$-smooth).**
> For any learning rate $0 < \eta \le 1/(4\bar{L})$, Theorem 6.9 of [1] gives:
> $$
> \mathbb{E}\left[\hat{U}\_n^g(\pi\_{\theta\_n^{\mathrm{opt}}}) -
> \hat{U}\_n^g(\pi\_{\bar{\theta}\_T})\right] \le  \frac{||\theta\_0 - \theta\_n^{\mathrm{opt}}||^2}{\eta T} + \frac{8\eta G_{\max}^2 H^2}{b}.
> $$
> where $\theta\_n^{\text{opt}} \in \arg\max\_\theta \hat{U}\_n^g(\pi\_\theta)$ is an optimal parameter and $\bar{\theta}_T$ is the average of the iterates. Finally, setting $\eta \propto 1/\sqrt{T}$ yields a **$\mathcal{O}(1/\sqrt{T})$** rate.
>
>
> * **PWLL with $\ell_2$ regularization (strongly concave and $\bar{L}$-smooth).** For the strongly concave objective
> $$
> \tilde{U}\_n^\lambda(\theta)=\hat{U}\_n^g(\pi\_\theta)-\tfrac{\lambda}{2}||\theta||^2,
> $$
> Theorem 6.12 of [1] ensures that for  $0<\eta \le 1/(2(\bar{L}+\lambda))$:
> $$
> \mathbb{E}\left[||\theta\_t - \theta_{n,\lambda}^{\mathrm{opt}}||^2\right] \le (1-\eta\lambda)^t ||\theta\_0 - \theta_{n,\lambda}^{\mathrm{opt}}||^2 + \frac{8\eta G\_{\max}^2 H^2}{\lambda b}.
> $$
> where $\theta\_{n,\lambda}^{\text{opt}} = \arg\max\_\theta \tilde{U}\_n^\lambda(\theta)$ is the unique optimal parameter.
>
>
> Thus, PWLL with regularization enjoys faster convergence. See Appendix E for more details. Note that these guarantees rely on PWLL’s concavity and smoothness; OPE objectives are non-concave and do not admit analogous global guarantees.
>
>
> [1] Garrigos, G., & Gower, R. M. (2023). Handbook of convergence theorems for (stochastic) gradient methods. arXiv preprint arXiv:2301.11235.

---

> > ### Author Response · Authors · 2025-11-23
> > **Part II**
> >
> > ### **3. Estimation–optimization trade-off: When does OPE beat PWLL?**
> >
> > The classical Bottou–Bousquet [2] excess risk decomposition can be applied on the suboptimality to understand the OPE-PWLL trade-off. Any learned policy's suboptimality decomposes as:
> >
> > $$V(\pi^\star) - V(\hat{\pi}) = \underbrace{V(\pi^\star) - V(\pi^{\Pi})}\_{\text{approximation}} + \underbrace{V(\pi^{\Pi}) - V(\pi^{\text{emp}})}\_{\text{estimation}} + \underbrace{V(\pi^{\text{emp}}) - V(\hat{\pi})}\_{\text{optimization}}$$
> >
> > where:
> > - $\pi^{\Pi}$ denotes the best policy within our parametric class
> > - $\pi^{\text{emp}} = \arg\max_{\pi \in \Pi} \hat{V}_n(\pi)$ is the empirical maximizer of the chosen objective
> > - $\hat{\pi}$ is the policy actually found by the optimizer
> >
> > Each component exhibits distinct behavior across methods:
> >
> > * **Approximation error**: Determined by the parametric class (linear-softmax): identical for both OPE and PWLL
> > * **Estimation error**: Statistical deviation from the population objective: OPE has a lower estimation error.
> > * **Optimization error**: Gap due to incomplete optimization: PWLL has a lower optimization error.
> >
> >
> > OPE outperforms PWLL when its estimation advantage outweighs its optimization disadvantage. Figure 13 in Appendix F.6 demonstrates this precisely: at $K=100$, both methods achieve comparable performance, with IPS (OPE) performing competitively. However, as $K$ increases, IPS exhibits severe optimization instability (validation reward collapses dramatically at $K=5000$) while cLPL (PWLL) maintains stable convergence across all action space sizes. This empirical pattern confirms that optimization challenges overwhelm OPE's estimation advantages in large-$K$ settings.
> >
> > [2] Bottou, Léon, and Olivier Bousquet. "The tradeoffs of large scale learning." Advances in neural information processing systems 20 (2007).
> >
> >
> > ---
> >
> > ### **4. Pessimistic / PAC-based OPL methods**
> >
> > We added new experiments comparing PWLL to pessimistic approaches (Figure 16, Appendix F.9). PWLL outperforms them, and pessimism, when using a reasonable regularization coefficient, does not resolve the inherent optimization bottleneck of OPE objectives.
> >
> > ---
> >
> >
> >
> > ### **5. Extension to offline RL**
> >
> > In RL, the same optimization challenges remain and are likely exacerbated. Our perspective suggests that the instability often observed in RL may partly stem from such landscape issues, alongside other factors established in the literature. Furthermore, our PWLL framework offers an optimization-based justification for reward/advantage-weighted behavioral cloning objectives, complementing their usual KL-regularized / trust-region interpretations. Rigorously extending this analysis to multi-step RL is left for future work.
> >
> > ---
> >
> >
> > ### **6. Robustness to misspecified rewards and deficient support**
> >
> > We additionally include experiments evaluating robustness to noisy reward feedback (Figure 11, Appendix F.4). PWLL objectives are affected by reward noise, but they remain stable and continue to outperform OPE objectives even under noisy rewards.
> >
> > We evaluate how expanding or restricting the logging-policy support affects policy learning, comparing PWLL with OPE objectives. Figure 15 in Appendix F.8 shows that PWLL consistently outperforms OPE methods across different support sizes. Here, $k_{\text{log}}$ denotes the logging support size, chosen to include high-reward actions (identified via SVD embeddings).
> >
> > ### **7. Clarification of asymptotic notation in Eq. (10)**
> >
> > > *"What does the asymptotic regime mean in Eq. (10)? Doesn't $\hat{r}$ become $r$ as $n \to \infty$?"*
> >
> > The reward model $\hat{r}$ is trained on a separate dataset (size $n_r$) before policy learning begins (dataset size $n$). This separation is standard in both theory where doubly robust methods require independent data for unbiasedness, and practice where reward models are pre-trained on separate data.
> >
> > Consequently, when we take $n \to \infty$ in our asymptotic analysis, $\hat{r}$ remains fixed: it's a pre-trained function independent of $n$. In fact, even if one used infinite data to learn $\hat{r}$, convergence to the reward function $r$ isn't guaranteed: reward models are often inherently biased due to model misspecification and fail to recover the true reward even asymptotically.
> >
> > We sincerely appreciate your thorough and constructive review. We hope the additional results and clarifications fully resolve your concerns, and we would greatly appreciate your consideration in raising your support for our paper.

---

### Official Review · Reviewer_mJqf · 2025-10-31

**Soundness:** 3
**Presentation:** 3
**Contribution:** 3
**Rating:** 6
**Confidence:** 3

**Summary:**

This paper argues that in offline contextual bandits with very large discrete action spaces, the dominant “estimator-centric” approach to off-policy learning ignores a crucial practical issue: the optimization landscape is hard to optimize. The authors make 4 main claims:
1.	Many existing OPE objectives induce highly non-concave objectives when paired with common parametric policies; they show that this pathology worsens as the size of action space grows.
2.	 When the underlying policy is parameterized as a full softmax over all actions, gradient computation and optimization remains costly and poorly conditioned (except POTEC, because it explicitly reduces the effective action space by fixing intra-cluster behavior).
3.	Policy-Weighted Log-Likelihoods (PWLLs) are easy to optimize because of their concavity under common parameterizations.
4.	Experimental results show PWLL methods are more robust than estimator-centric baselines across hyperparameters, and the gap between PWLL and previous work widens as the action space grows.

**Strengths:**

The paper shows clearly how estimator-centric OPE objectives break down in very large action spaces and how it can be beneficial to analyze the OPE problem from an optimization perspective. Theoretically, they derive how different objectives behave in the infinite-data regime. Then switching to log-likelihood (PWLL) objectives, combined with linear softmax parameterizations, yields favorable curvature (concavity) properties and improved optimization behavior in practice. Empirically, they validate these insights with extensive large-scale experiments on real recommendation datasets, and the performance gap widens as the action space becomes larger.
Taken together, these theoretical and empirical contributions make a convincing case for an optimization-centric approach; the work appears to be among the first to systematically demonstrate, both analytically and at scale, that this reframing improves training stability and policy quality.

**Weaknesses:**

- Missing baselines, e.g.  Contextual Bandits with Large Action Spaces: Made Practical (Zhu et, al. ICML 2022)
- Reproducibility & ablations: while the datasets are large and convincing, the paper would benefit from more ablations: (i) seed variability and statistical significance, (ii) whether PWLL remains robust to different hyperparameters (e.g. $\beta, \tau$) (iii) how PWLL performs in comparison to POTEC with different cluster size.
- Lack of qualitative results e.g. the optimization landscape in toy examples

**Questions:**

1. Clustering sensitivity: POTEC and MIPS rely on embeddings/clustering. Could the authors include sensitivity analysis to embedding quality or cluster size? How often does poor clustering make POTEC fail?
2. Under what conditions does the bias introduced by PWLL become severe, and how should practitioners mitigate it?
3. Would the argument (favor optimization over estimation) still hold for non-trivial (multilayer) softmax policies?

---

> ### Author Response · Authors · 2025-11-23
>
> Thank you for the detailed and positive assessment. We appreciate that you found the optimization-focused perspective compelling. Below, we address your concerns and list the concrete updates made in the revised version of our paper.
>
> ### **Summary of changes**
>
> * Added a discussion of Zhu et al. (ICML 2022) and other **online** large-action bandit work, clarifying its distinction from our **offline** setting (Appendix A).
> * Reported **multi-seed** and **more ablation** results (Appendix F).
> * Added a **2D visualization** for OPE and PWLL optimization landscapes (Figure 1 and lines 340-352).
> * Clarified how our analysis and insights extend to deeper networks (Remark 3.2 in lines 354-360).
>
> ---
>
> ### **1) Reproducibility, ablations, and qualitative results**
>
> **Seeds.** We added multi-seed results (mean ± std). In Figures 9 and 10 in Appendix F.3, PWLL and objective-aware parametrizations reduce variance and keep performance consistently robust.
>
> **Hyperparameters.** We vary PWLL-specific hyperparameters ($\tau$, $\beta$) in Figure 12 (Appendix F.5). PWLL performance changes smoothly across all ranges and remains superior to OPE objectives throughout. In practice, optimization error dominates any bias introduced by suboptimal hyperparameter choices.
>
> **Cluster size (MIPS/POTEC).** Figure 14 (Appendix F.7) shows cluster-size ablations: performance degrades when clusters are too small, especially on the larger-$K$ Twitch dataset, because this effectively reintroduces large action spaces and associated instability.
>
> **Toy landscapes.** Figure 1 (and its description in lines 340–352) presents a 2D toy example illustrating OPE’s landscape pathologies versus the single smooth optimum of PWLL. These issues become more severe in large-$K$ settings, although they cannot be directly visualized there.
>
> ---
>
> ### **2) PWLL bias: when it matters and how to mitigate it**
>
> We distinguish between two types of bias:
> * **Estimation bias:** Deviation between the expected objective and true value: $\mathrm{Bias}_{\text{est}}(\pi) = \mathbb{E}[\hat U_n^{g}(\pi)] - V(\pi)$
> * **Learning bias:** Difference between the maximizers of the expected objective (i.e., asymptotic solution) and the optimal policy: $\pi_*^{g} = \arg\max_{\pi} \mathbb{E}[\hat U_n^{g}(\pi)]$ vs. $\pi_*= \arg\max_{\pi} V(\pi)$
>
> Since we focus on **policy learning**, not value estimation, we only considered learning bias (asymptotic solutions) in the paper. Now, regarding the estimation bias:
>
> **Estimation bias is high by design.** PWLL is **not** a value estimator. Thus, it naturally has high estimation bias. The magnitude depends on the choice of $g$. For cLPI, the bias is:
>
> $$\mathrm{Bias}\_{\text{est}}^{\text{cLPI}}(\pi) = \mathbb{E}\_X \left[ \sum\_{a \in \mathcal{A}} r(X,a)\left[\frac{\pi\_0(a|X)}{\max( \pi\_0(a|X),\tau)}\log \pi(a|X) - \pi(a|X)\right]\right]$$
>
> Here, the absolute bias becomes large for high-reward actions $a'$ for which coverage is misaligned: either $\pi_0(a'|x)$ is large but $\pi(a'|x)$ is tiny, or vice versa.
>
> However, what actually matters is **learning bias**, which is the quality of the asymptotic solution. Each $g$ induces a specific, interpretable asymptotic solution. For example, cLPI (PWLL objective) has the **same asymptotic solution** as cIPS (OPE estimator). The difference is that PWLL achieves these solutions **much more stably** due to its benign optimization landscape, explaining superior finite-sample performance.
>
> ---
>
> ### **3) Beyond linear softmax policies**
>
> * **Frozen (pre-trained) encoder + trainable linear head** (used in recommender systems): All guarantees hold since we only optimize the linear head parameters.
>
> * **Fully trainable deep networks**: Global concavity is lost for any non-trivial architecture. However, PWLL provides a structural advantage: it replaces linear-in-$\pi$ terms with $\log \pi$, yielding familiar log-likelihood gradients:
> $$\nabla\_\theta \hat{U}\_n^{g}(\pi\_\theta) \propto g(\cdot) \cdot \nabla\_\theta \log \pi\_\theta(A|X)$$
> Log-softmax gradients are well-scaled and avoid saturation: the same property underlying cross-entropy's success in deep learning. Full analysis of deep policies is left for future work.
>
> ---
>
> ### **4) Missing baselines**
>
> Zhu et al. (2022) study *online* contextual bandits, whereas we focus on the offline setting. We now explicitly discuss Zhu et al. (2022) and related online methods in the extended related work section to clarify this distinction.
>
> ---
>
> Thank you again for the detailed and encouraging review. We hope the clarifications and new experiments resolve your concerns and strengthen your confidence in our submission. We would greatly appreciate your consideration in raising your support for our paper.

---

### Official Review · Reviewer_hky7 · 2025-11-03

**Soundness:** 2
**Presentation:** 2
**Contribution:** 2
**Rating:** 4
**Confidence:** 2

**Summary:**

The authors propose a policy-weighted log-likelihood (PWLL) objective for off-policy learning. They present various formulations of off-policy learning objectives and show that their proposed equation generalizes previous objective functions while the policy probability model is replaced by its logarithmic form. When applying the LPI, cLPI, and RegKL specializations, they demonstrated that their objective function is easily optimized, although the resulting solution can be potentially biased.

**Strengths:**

The authors proposed a generalized formulation that integrates and extends previous off-policy learning objectives whose optimization is easily performed. The method is novel.

**Weaknesses:**

The authors provided empirical evidence showing that the proposed method outperforms existing approaches. Nonetheless, it is difficult to recommend the paper for acceptance due to the following deficiencies. The motivation for introducing the logarithmic form is unclear, and the authors fail to adequately justify the modification either mathematically or intuitively.

**Questions:**

It remains unclear whether the introduction of logarithmic term is theoretically or intuitively well-motivated and whether it avoids pathological cases while achieving an optimal solution comparable to that of previous methods.

---

> ### Author Response · Authors · 2025-11-23
>
> We appreciate your review and your recognition of our paper’s novelty. Your primary concern lies in the motivation for the logarithmic term in PWLL. Below, we summarize the key revisions made to the paper and provide detailed clarifications that directly address this point.
>
> ### **Summary of changes**
> Based on your feedback, we made the following improvements:
>
> * Clarified the motivation and intuition behind PWLL in Section 3 (lines 297-302 and lines 313-338).
> * Added an ablation study in Appendix F.5 (Figure 12 (b)) comparing the use of the logarithmic term versus its removal.
>
> ---
>
> ### **1). Theoretical motivation (concavity)**
>
> * **Props. 2.1–2.2** showed that objectives that are linear in $\pi_\theta$ suffer from flat regions and local traps because the softmax can be highly non-concave even for linear scores . Therefore, we wanted to avoid such linearity in $\pi_\theta$.
>
>
> * Luckily, the map $s \mapsto \log \mathrm{softmax}(s)$ is concave. Thus, for linear softmax policies, the composition $\log \pi_\theta = \log \mathrm{softmax}(s_\theta)$ remains concave in $\theta$.
>
>
> * This guarantees a global maximum and efficient optimization, formalized in **Prop. 3.1**.
>
> ---
>
> ### **2). Intuitive motivation (weighted MLE)**
> Beyond the concavity argument, PWLL has a simple intuition: it turns off-policy learning into weighted maximum-likelihood estimation, where each logged action gets more or less weight depending on how useful it is for learning a good policy.
>
> In the PWLL objective: \$\hat{U}\_n^g(\pi) = \frac{1}{n}\sum_{i=1}^n g(R_i,\pi_0(A_i|X_i))\log \pi(A_i|X_i)\$:
>
> * The $\log \pi$ term performs standard MLE
> * The weight $g$ determines which samples are *desirable* for learning, and different choices of $g$ encode different definitions of desirability. For example:
>
>   - **cLPI** $g(r, \pi_0(a|x)) = \frac{r}{\max(\pi_0(a|x), \tau)}$:
>
>     - Desirable = high-reward + rarely-tried (up to threshold $\tau$ for stability)
>
>     - Rationale is to learn most from successful exploration: actions that worked well despite low logging probability (up to a certain threshold $\tau$ for stability).
>
> ---
>
> ### **3). Evidence (pathologies avoided)**
>
> * **Pathologies Avoided:** The log term's concavity (**Prop. 3.1**) is precisely what allows us to avoid the non-log pathologies (**Props. 2.1–2.2**).
>
> * **No-Log Ablation:** In **Appendix F.5 (Figure 12 (b))**, replacing $\log \pi$ with $\pi$ in RegKL (a PWLL method) significantly degrades optimization and performance for varying values of hyperparameters $\beta$.
>
> * **Optimality Preserved:** PWLL specializations share the *same asymptotic solution* as non-log counterparts, but reach it more reliably in practice due to the benign optimization landscape.
>
> ---
>
> ### **Answers to your questions**
>
> > Is the logarithmic term theoretically or intuitively well-motivated?
>
> **Yes.** see **1)** and **2)** above.
>
> > Does it avoid pathological cases while achieving an optimal solution comparable to previous methods?
>
> **Yes.** see **3)** above.
>
> We hope this detailed clarification fully addresses your concern about the log term's motivation. Given this was the only concern mentioned in the review, we would greatly appreciate your consideration for raising your support of our paper.

---

### Author Response · Authors · 2025-11-30
**Summary of Revisions and Response to Reviewers**

We welcome the new Area Chair and deeply appreciate your time and consideration. We summarize below our contributions and how the revision resolves reviewer concerns.

### **I) Short main claim.**
In off-policy learning with large action spaces, **optimization error dominates estimation error**, acting as the primary performance bottleneck.

### **II) Detailed main claim.**
* **Problem:** We prove that OPE objectives (IPS, DR, etc.), including those tailored for large action spaces (MIPS, PC, etc.), induce pathological landscapes characterized by extensive plateaus and local maxima, and that these issues worsen as the action space grows (**Props. 2.1 & 2.2**).
* **Partial Solution:** We introduce **objective-aware policy parametrization**, which restricts the effective action space (actions assigned non-zero probability) based on estimator bias. This is useful for practitioners who must work with OPE objectives as it mitigates instability since the search space is restricted. However, this does not solve the underlying non-concavity.
* **Full solution:** We propose **policy-weighted log-likelihood (PWLL)**, which is **globally concave** (**Prop. 3.1**) while sharing the same **asymptotic solution** (infinite-data optima) as OPE estimators, but with much more easier optimization.

Both solutions substantially outperform state-of-the-art, complex OPE objectives in large action spaces (**Sec. 4**).

### **III) Evidence in the paper.**
* **Theory.** We prove that OPE objectives (including those tailored for large action spaces) suffer from optimization pathologies (vanishing gradients, local optima) that scale with $K$ (**Props. 2.1–2.2**), whereas PWLL guarantees global concavity (**Prop. 3.1**).
* **Experiments.** Across real-world benchmarks with up to **1M actions**, PWLL consistently outperforms sophisticated OPE baselines with superior training stability.


### **IV) Supporting evidence in the rebuttal**

We added new results that directly address the reviewers’ concerns:

* **1. Explicit optimization guarantees (App. E).** We prove that PWLL is $L$-smooth with bounded gradient variance, yielding global convergence guarantees that non-concave OPE objectives do not enjoy.
* **2. Landscape visualization (Fig. 1).** 2D visualizations explicitly contrast the pathological landscapes of OPE (plateaus and spikes) with the smooth, globally concave landscape of PWLL.
* **3. Extensive ablations (App. F).**

  * **Multi-seed (Figs. 9–10):** PWLL and objective-aware parametrization exhibit much lower variance across randomization seeds than OPE objectives.
  * **Log-term importance (Fig. 12b):** Removing the log term destroys the optimization gains, confirming its necessity for good performance.
  * **Robustness (Figs. 11 to 15):** PWLL consistently dominates baselines under varying reward noise, hyperparameters, number of actions, cluster sizes, and logging-support sizes.
* **4. Pessimism baseline (Fig. 16).** Adding pessimism (e.g., regularization toward $\pi_0$) to OPE objectives does not resolve their optimization pathologies; PWLL remains superior.

### **V) Summary of reviewer main concerns vs. our fixes**
| Reviewer | Main Concern | Status in Revised PDF |
| :--- | :--- | :--- |
| **R-hky7** | Motivation for the log term.                                                      | **Resolved:** Added intuition and a no-log ablation showing degraded optimization (Fig. 12b, App. F.5).                                                                                                                                                                                    |
| **R-mJqf** | Missing baseline; need ablations, toy visualization, deep-policy discussion.      | **Resolved:** Clarified that the requested baseline is an online algorithm (not offline) and discussed it in extended related work (App. A); added multi-seed experiments and extensive ablations (App. F); included 2D visualizations (Fig. 1); and clarified the extension to deep policies (Rem. 3.2). |
| **R-Vgm9** | Optimization theory, novelty and RL connection, pessimism, estimation–optimization trade-off, robustness. | **Resolved:** Added optimization convergence guarantees (App. E); clarified novelty and implications for RL; added pessimistic baselines (Fig. 16); explicitly illustrated the estimation–optimization trade-off via the Bottou–Bousquet decomposition and small-$K$ experiments (Fig. 13); and added extensive ablations for robustness (App. F).                     |
| **R-u2kL** | Deep-policy extension; generalization bounds; reward sensitivity.                 | **Resolved:** Added extensive ablations ablations (App. F); clarified the deep-policy extension (Rem. 3.2); and explained that standard generalization bounds are attainable under mild assumptions.|

In summary, the rebuttal provides clarifications and confirmatory results that reinforce the paper’s original claims and evidence. We are grateful to the committee for their time and consideration.

---

### Meta-Review · Area_Chair_Yrkz · 2026-01-13

**Summary:**

The paper investigates off-policy learning (OPL) of offline contextual bandit in large action spaces, arguing that optimization difficulties (non-concavity) are a more significant bottleneck than estimation errors. It presensts Policy Weighted Log-Likelihood (PWLL) objectives, as optimization-friendly alternatives to standard OPE estimators. While the paper provides empirical analysis and visualizations of optimization landscapes, the core algorithmic contribution is limited.

**Reviewer Concerns:**

**Concerns addressed by rebuttal:**
- Issues regarding the theoretical motivation for the logarithmic term and the convergence guarantees of the proposed objectives were resolved during the rebuttal phase.


**Outstanding concerns**
- The proposed methods (e.g., RegKL, cLPI) are mathematically equivalent to reward-weighted regression (RWR) or the bandit counterparts of established offline RL algorithms such as CRR/AWAC/AWR. Although cLPI is presented as a new variant, its empirical results (Figure 12) show its performance is indistinguishable from RegKL, which is effectively a standard RWR/CRR implementation. The fundamental idea of "using weighted log-likelihood to avoid the instability of direct value maximization (weighted BC)" is a well-established principle in the broader offline RL literature.
- The paper frames the optimization vs estimation trade-off as a novel discovery. However, this trade-off is precisely why advantage-weighted methods became standard in offline RL. By applying these known techniques to contextual bandits without explicitly positioning them as adaptations of existing RL methods, the paper overstates its novelty. It essentially solves the problem by applying a standard solution from a neighboring field, which constitutes an incremental application rather than a fundamental methodological advance.

**Reviewer Scores:**

- Reviewer hky7 (6): Initially positive about intuition, but the lack of novelty undermines the fair contribution rating.
- Reviewer mJqf (4): Correctly identified the missing baselines and the similarity to existing work.
- Reviewer Vgm9 (4): Raised valid concerns about the limited novelty in algorithmic form, which remains the primary bottleneck.
- Reviewer u2kL (6): Appreciated the analysis but acknowledged the limitations regarding deep policies.

---

### Decision · Program_Chairs · 2026-01-26

Reject